# SEARCH OR THINK? RETHINKING ITERATIVE RAG FROM AN ENTROPY PERSPECTIVE

## ABSTRACT

Retrieval-Augmented Generation (RAG) has emerged as a powerful paradigm for Large Language Models (LLMs) to address knowledge-intensive queries requiring domain-specific or up-to-date information. To handle complex multi-hop questions that are challenging for single-step retrieval, iterative RAG approaches incorporating reinforcement learning have been proposed. However, existing iterative RAG systems typically *think first* to decompose questions without leveraging information about the available retrieval corpus, leading to inefficient retrieval and reasoning chains that cascade into suboptimal performance. In this paper, we introduce Search-Initialized Thinking (SIT), a novel framework that *searches first* before think in iterative RAG systems with contextually relevant retrieved knowledge. From an entropy perspective, we demonstrate that incorporating initial knowledge with search reduces unnecessary exploration during the reasoning process, enabling the model to focus more effectively on relevant information subsets. Extensive experiments on six standard RAG datasets demonstrate that by establishing a stronger reasoning foundation, SIT significantly improves retrieval precision, reduces cascading errors, and enhances both performance and efficiency. Moreover, SIT proves effective as a versatile, training-free inference strategy that scales seamlessly to large models.Generalization tests across diverse datasets and retrieval corpora confirm the robustness of our approach. Overall, SIT advances the state-of-the-art in iterative RAG systems while illuminating the critical interplay between structured reasoning and efficient exploration in reinforcement learning-augmented frameworks.

## 1 INTRODUCTION

Large Language Models (LLMs) have demonstrated remarkable capabilities in natural language understanding and generation, yet they face fundamental limitations when dealing with knowledge-intensive tasks that require access to up-to-date or domain-specific information. Retrieval-Augmented Generation (RAG) has emerged as a promising paradigm to address these limitations by dynamically incorporating external knowledge from retrieval corpora into the generation process (Karpukhin et al., 2020; Lewis et al., 2020). Standard RAG systems perform a single retrieval step followed by generation, but the intrinsic difficulty of retrieving multi-hop information in one step causes a lot of failure. Recent advances have shown that iterative approaches where models can perform multiple rounds of retrieval and reasoning—significantly improve performance on complex multi-hop reasoning tasks(Jin et al., 2025a; Guan et al., 2025; Luo et al., 2025a; Song et al., 2025). However, although assumed well, these iterative systems can still suffer from retrieval failure, resulting from the plan failure which leads to the suboptimal reasoning chains, particularly when the initial reasoning step lacks sufficient contextual grounding. These scenarios are illustrated in Figure 1 with a real example from the dataset.

Iterative RAG systems(Jin et al., 2025a; Song et al., 2025) are often optimized by Reinforcement Learning (RL)(Schulman et al., 2017a; Shao et al., 2024b), offering a principled approach to learn effective retrieval and reasoning strategies. RL-based RAG frameworks treat the retrieval and generation process as a sequential decision-making problem, where agents learn to search for information and generate responses to maximize cumulative rewards based on answer accuracy and efficiency metrics. The success of RL training heavily depends on the quality of the exploitation and the exploration efficiency during the learning process. Recent studies on entropy(Wang et al., 2025;

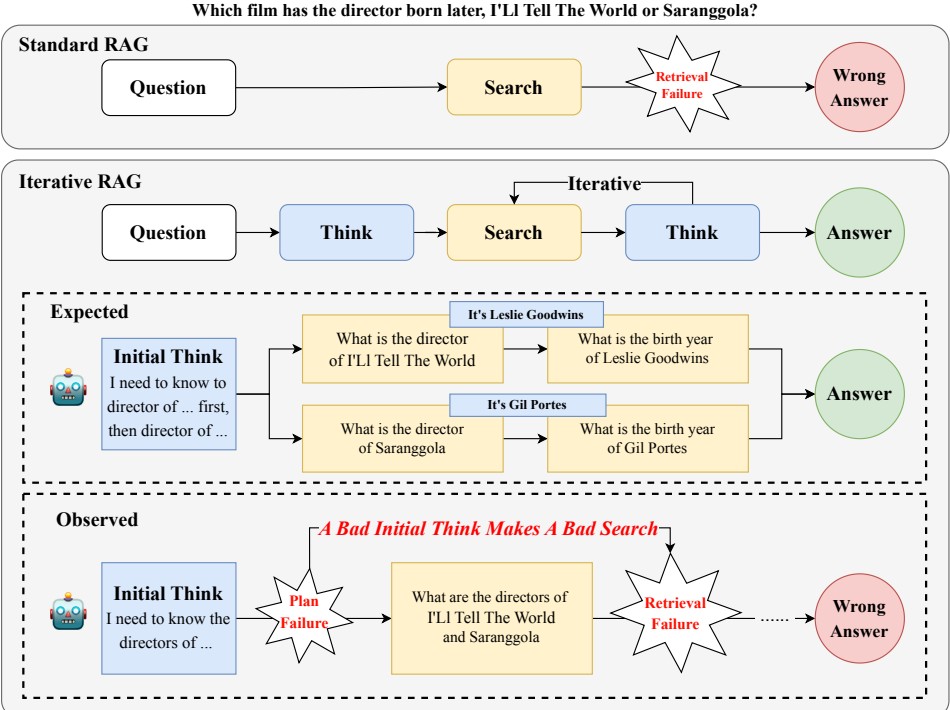

Figure 1: Standard RAG and Iterative RAG pipeline. While standard RAG suffers from the impossibility of multi-hop retrieval in one step, iterative RAG also suffers from plan failure in the initial think, which is caused by lack of information of the retrieval set.

Cui et al., 2025) show that entropy measurement is a good signal for this exploitation and exploration balance, which is important because the exploitation of retrieved information and exploration in the retrieval set control the whole reasoning process. Poor initial reasoning steps in exploration can lead to compounding errors throughout the iterative process.

From both the perspective of an iterative RAG system and the RL training dynamics, the quality of initial thinking plays a crucial role in the effectiveness of generating right answers. When models begin their reasoning process without adequate contextual knowledge, they often generate misguided hypotheses or pursue irrelevant reasoning paths relying on themselves, as called *Model-Initialized Thinking*, which is far from the information the environment can give, leading to a cascade of poor retrieval decisions and incorrect conclusions. This problem is particularly pronounced in the early stages of RL training, where random or poorly informed initial actions can significantly hinder the learning process. By enhancing the initial thinking step with relevant retrieval, we hypothesize that models can establish more accurate reasoning foundations, leading to better exploration strategies with less entropy and more efficient learning dynamics. This **Search-Initialized Thinking (SIT)** not only improves the immediate reasoning quality but also provides clearer learning signals for the RL algorithm, enabling faster roads to the right answer.

Our contribution is as follows:

- **Search-Initialized Thinking (SIT).** We propose a novel approach that augments the initial thinking step in iterative RAG systems with retrieved contextual knowledge, providing models with better grounding before entering the RL-optimized iterative retrieval and generation process. This framework significantly improves the quality of reasoning foundations and reduces the likelihood of cascading errors in subsequent iterations.
- **Analysis from an Entropy Perspective.** We analyze the training dynamics of Group Relative Policy Optimization (GRPO)(Shao et al., 2024b) in iterative RAG from an entropy perspective and show that with lower entropy in the training phase, instead of insufficient exploration, our approach leads to more efficient exploration strategies focusing on the retrieval set, faster roads to the answer during RL training compared to traditional approaches that start with uninformed, model initialized thinking.

- **Comprehensive Experimental Validation.** We conduct extensive experiments on standard RAG datasets, showing consistent improvements in both answer accuracy and retrieval recall. Besides, generalization experiments show no degrading of generalization with our method.

## 2 RELATED WORKS

### 2.1 RETRIEVAL-AUGMENTED GENERATION

The concept of augmenting language models with external knowledge retrieval has gained significant traction in recent years. Early work by (Karpukhin et al., 2020) introduced Dense Passage Retrieval (DPR), which demonstrated the effectiveness of dense vector representations for retrieval in open-domain question answering. (Lewis et al., 2020) proposed Retrieval-Augmented Generation and a lot of works(Gao et al., 2023; Li et al., 2023) has emerged. To apply better retrieval, LightRAG(Guo et al., 2025) employs a dual-level retrieval system for better generation. Structure-based retrieval methods like GraphRAG(Edge et al., 2025), PathRAG(Chen et al., 2025), HippoRAG2(Gutiérrez et al., 2025), HyperGraphRAG(Luo et al., 2025b) have been proposed to utilize fine-grained retrieval like entities or links and generate better responses. Traditional single-step RAG systems often fall short when dealing with complex reasoning tasks that require multiple pieces of evidence. This limitation has motivated research into iterative RAG systems.

### 2.2 ITERATIVE AND MULTI-HOP RAG APPROACHES

Chain-of-Thought (CoT) prompting (Wei et al., 2022) encourages models to generate intermediate reasoning steps, effectively simulating an iterative thinking process. IRCoT (Trivedi et al., 2022b) demonstrated that interleaving retrieval and generation steps can significantly improve performance on multi-hop reasoning tasks. ITER-RETGEN (Shao et al., 2023) proposed a framework where models can decide when to retrieve additional information based on their confidence levels. WebGPT (Nakano et al., 2021) showed that models can be trained to browse the web iteratively to gather information for answering questions. ReAct (Yao et al., 2023) combined reasoning and acting in language models, enabling them to perform dynamic retrieval based on their reasoning traces. More recent work by (Jiang et al., 2023a) introduced Self-RAG, which uses reflection tokens to control retrieval timing and assess the quality of retrieved passages,while Self-ask, proposed by (Press et al., 2023), implements an autonomous question formulation mechanism during the reasoning process. FLARE (Jiang et al., 2023b) incorporates adaptive retrieval when LLMs generate low-confidence tokens.

### 2.3 REINFORCEMENT LEARNING FOR RAG OPTIMIZATION

The application of reinforcement learning to optimize RAG systems has emerged as a promising research direction. Several approaches, such as R1-Searcher(Song et al., 2025), R3-RAG(Li et al., 2025b), and DeepRAG(Guan et al., 2025), employ a two-stage training process. They first use manually curated data to perform Supervised Fine-Tuning (SFT) on the LLM, and subsequently apply reinforcement learning to further align the model with the available knowledge boundaries.Similarly, s3 (Jiang et al., 2025) proposes a modular framework that employs RL to optimize a search agent while keeping the generator frozen, focusing on input context optimization rather than joint reasoning. A critical problem is that some multi-hop questions have more than one good reasoning paths, which requires high quality for sft data. Search-R1(Jin et al., 2025a) and Graph-R1(Luo et al., 2025a) directly applies reinforcement learning on LLMs. Consequently, these approaches rely more heavily on the LLM's innate reasoning capabilities to solve the questions without a preceding SFT stage. This may introduce redundant paths when LLM does not align with the retrieval set. Our method applies searching initialized think to alleviate this problem.

## 3 PRELIMINARIES

### 3.1 PPO

Proximal Policy Optimization (PPO) (Schulman et al., 2017b) is an actor-critic reinforcement learning algorithm that has become the predominant method for RL fine-tuning of large language models

(Ouyang et al., 2022). For language model fine-tuning, PPO maximizes the following objective:

$$\mathcal{J}_{PPO}(\theta) = \mathbb{E}_{[q \sim P(Q), o \sim \pi_{\theta_{old}}(O|q)]} \left[ \frac{1}{|o|} \sum_{t=1}^{|o|} \min\left(r_t(\theta) A_t, \text{clip}(r_t(\theta), 1-\epsilon, 1+\epsilon) A_t\right) \right], \quad (1)$$

where $r_t(\theta) = \frac{\pi_\theta(o_t|q, o_{<t})}{\pi_{\theta_{old}}(o_t|q, o_{<t})}$ is the probability ratio between the current policy $\pi_\theta$ and the old policy $\pi_{\theta_{old}}$. Here, $q$ and $o$ represent questions sampled from the dataset $P(Q)$ and corresponding outputs generated by the old policy, respectively. The clipping parameter $\epsilon$ constrains the policy ratio to the interval $[1-\epsilon, 1+\epsilon]$, preventing destabilizing updates. $A_t$ denotes the advantage function, typically computed using Generalized Advantage Estimation (GAE)(Schulman et al., 2015) based on rewards and a learned value function $V_\psi$.

## 3.2 GRPO

(Shao et al., 2024b) propose Group Relative Policy Optimization (GRPO), illustrated in Figure 2. GRPO eliminates the need for value function approximation by using the average reward of multiple sampled outputs as a baseline. For each question $q$, GRPO samples a group of $G$ outputs $\{o_1, o_2, \ldots, o_G\}$ from the old policy $\pi_{\theta_{old}}$ and optimizes the following objective:

$$\mathcal{J}_{GRPO}(\theta) = \mathbb{E}_{[q \sim P(Q), \{o_i\}_{i=1}^G \sim \pi_{\theta_{old}}(O|q)]}$$
$$\left[ \frac{1}{G} \sum_{i=1}^{G} \frac{1}{|o_i|} \sum_{t=1}^{|o_i|} \min\left( r_{i,t}(\theta) \hat{A}_{i,t}, \text{clip}(r_{i,t}(\theta), 1-\epsilon, 1+\epsilon) \hat{A}_{i,t} \right) - \beta \mathbb{D}_{KL}[\pi_\theta || \pi_{ref}] \right],$$
$$(2)$$

where $r_{i,t}(\theta) = \frac{\pi_\theta(o_{i,t}|q, o_{i,<t})}{\pi_{\theta_{old}}(o_{i,t}|q, o_{i,<t})}$ is the probability ratio, and $\hat{A}_{i,t}$ represents the advantage computed using relative rewards within each group:

$$\hat{A}_{i,t} = \widetilde{r}_i = \frac{r_i - \text{mean}(\mathbf{r})}{\text{std}(\mathbf{r})} \quad (3)$$

where $\mathbf{r} = \{r_1, r_2, \cdots, r_G\}$ is the rewards tensor of $G$ samples in the group correspondingly. The group-relative advantage computation aligns naturally with how reward models are trained—on comparative datasets where outputs for the same question are ranked against each other.

## 4 METHOD

We propose **Search-Initialized Thinking (SIT)**, a novel approach that enhances iterative RAG systems by incorporating retrieved knowledge before the initial thinking step. Our method addresses the fundamental limitation of Model-Initialized Thinking, in all existing iterative RAG systems where models begin reasoning without sufficient contextual grounding, often leading to suboptimal retrieval strategies and redundant exploration during reinforcement learning.

Figure 2 illustrates the GRPO training pipeline of SIT. The policy LLM receives Initialized Knowledge $\mathcal{P}_0$ from the SearchEngine before its first thinking step. Subsequently, the model proceeds with the standard rollout and update phases as in conventional GRPO training. Algorithm is referred to Appendix B.

### 4.1 SEARCH INITIALIZATION

Given an input question $q$, our SIT approach first performs an initial retrieval step to gather relevant knowledge before generating the initial thinking step. Specifically, we retrieve the top-$k$ most relevant passages from the knowledge corpus $\mathcal{D}$ using a retriever:

$$\mathcal{P}_0 = \text{Retrieve}(q, \mathcal{D}, k), \quad (4)$$

where $\mathcal{P}_0 = \{p_1, p_2, \ldots, p_k\}$ represents the initially retrieved passages.

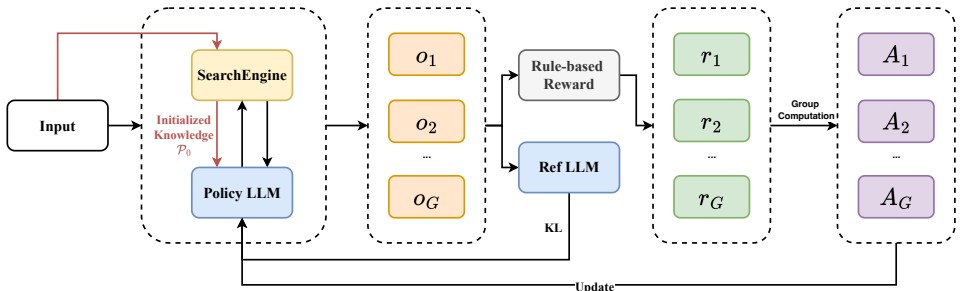

Figure 2: GRPO training with SIT.

## 4.2 ITERATIVE THINKING AND SEARCHING

Following the initial search, our method proceeds with iterative thinking and searching, now grounded by initial knowledge, until a final answer is generated. The action pipeline is set as $[a_0, a_1, a_2, ...a_t]$ where $a_0$ is **Search** and at each subsequent step $i > 0$, action $a_i$ is **Search** or **Answer** if $a_{i-1} =$ **Think** and **Think** if $a_{i-1}! =$ **Think**. Each action is defined as:

- **Think**: Generate reasoning steps based on current knowledge.
- **Search**: Query the knowledge base for additional information.
- **Answer**: Provide the final answer when sufficient information is gathered.

To guide the model in producing this sequence of actions, we employ the prompt detailed in Table 1, which instructs it to generate structured outputs.

Table 1: Template for the updated prompt. Note that initial knowledge is provided within `<knowledge>`...`</knowledge>` at the beginning, and additional retrieved knowledge is placed within the same tags after `</query>`.

| |
|---|
| Answer the given question. You can query from knowledge base provided to you to answer the question. You can query knowledge as many times as you want. The initial knowledge you need for the first think is between `<knowledge>`...`</knowledge>`. You must first conduct reasoning inside `<think>`...`</think>` relied on the initial knowledge. If you need to query knowledge, you can set a query statement between `<query>`...`</query>` to query from knowledge base after `<think>`...`</think>`. When you have the final answer, you can output the answer inside `<answer>`...`</answer>`. Question: question. `<knowledge>`Knowledge`</knowledge>`. Assistant: |

## 4.3 THEORETICAL ANALYSIS

In this section we propose the following proposition:

**Proposition 1.** Search-Initialized Thinking is better than Model-Initialized thinking in iterative RAG from an entropy perspective.

*Proof.* The formal proof is provided in Appendix D, and the empirical results regarding entropy are presented in Section 6.1. □

## 5 EXPERIMENTS

We choose two RAG methods based on reinforcement learning as our backbone, Search-R1(Jin et al., 2025a) and Graph-R1(Luo et al., 2025a), accompanied with two different dataset splitting, to show our method's robustness across different methods and retrieval set. In Search-R1 setting, models are trained in two IND (in-domain) datasets (HotpotQA and NQ) and other datasets are OOD (out-of-domain) datasets for test. In Graph-R1 setting, models are trained within each dataset. Furthermore, a comprehensive retrieval set with chunks using the full Wikipedia corpus (Fullwiki) is used in the Search-R1 setting, and a smaller, dataset-specific structure-augmented retreival set is

Table 2: Main results in Graph-R1 setting with best in **bold**. ⊙ means prompt engineering, ◉ means training, ⊘ means no knowledge interaction, ⊞ means chunk-based knowledge, and ⚒ means graph-based knowledge.

| Method | 2Wiki. | | HotpotQA | | Musique | | NQ | | PopQA | | TriviaQA | | Avg. | | |
|---|---|---|---|---|---|---|---|---|---|---|---|---|---|---|---|
| | EM | F1 | EM | F1 | EM | F1 | EM | F1 | EM | F1 | EM | F1 | EM | F1 | R-S |
| *GPT-4o-mini* | | | | | | | | | | | | | | | |
| ⊙ ⊘ NaiveGeneration | 4.69 | 17.03 | 18.75 | 31.79 | 3.13 | 11.45 | 2.34 | 21.59 | 10.36 | 25.95 | 28.91 | 47.73 | 11.36 | 25.92 | - |
| ⊙ ⊞ StandardRAG | 7.03 | 22.31 | 35.16 | 46.70 | 9.38 | 17.31 | 7.03 | 26.85 | 18.75 | 30.58 | 31.25 | 48.55 | 18.10 | 32.05 | 52.68 |
| ⊙ ⚒ GraphRAG | 3.91 | 16.02 | 19.53 | 31.67 | 7.03 | 15.14 | 3.91 | 20.31 | 8.59 | 20.92 | 32.03 | 45.13 | 12.50 | 24.87 | 32.48 |
| ⊙ ⚒ LightRAG | 3.13 | 16.59 | 18.75 | 30.70 | 3.91 | 14.39 | 2.34 | 19.09 | 5.47 | 24.47 | 25.00 | 40.18 | 9.77 | 24.24 | 47.42 |
| ⊙ ⚒ PathRAG | 3.91 | 12.42 | 10.94 | 23.12 | 3.13 | 11.49 | 2.34 | 20.01 | 2.34 | 15.65 | 19.53 | 37.44 | 7.03 | 20.02 | 46.71 |
| ⊙ ⚒ HippoRAG2 | 7.03 | 16.27 | 19.53 | 31.78 | 6.25 | 12.37 | 7.81 | 24.56 | 9.38 | 21.10 | 32.81 | 48.86 | 13.80 | 25.82 | 36.41 |
| ⊙ ⚒ HyperGraphRAG | 4.69 | 21.14 | 21.88 | 37.46 | 6.25 | 20.40 | 3.91 | 22.95 | 13.28 | 29.48 | 28.91 | 44.95 | 13.15 | 29.40 | 61.82 |
| *Qwen2.5-7B-Instruct* | | | | | | | | | | | | | | | |
| ⊙ ⊘ NaiveGeneration | 3.12 | 12.25 | 6.25 | 18.58 | 0.00 | 4.06 | 1.56 | 13.00 | 0.78 | 12.82 | 7.03 | 24.51 | 3.12 | 14.20 | - |
| ⊙ ⊞ StandardRAG | 7.81 | 12.75 | 10.16 | 21.10 | 0.78 | 4.53 | 1.56 | 15.97 | 3.12 | 13.10 | 8.59 | 24.90 | 5.34 | 15.39 | 52.67 |
| ◉ ⊘ SFT | 11.72 | 20.28 | 19.53 | 27.59 | 5.47 | 10.02 | 5.12 | 19.02 | 20.31 | 27.93 | 31.25 | 39.21 | 15.57 | 24.01 | - |
| ◉ ⊘ R1 | 25.00 | 30.99 | 31.25 | 37.05 | 7.03 | 14.53 | 16.41 | 28.45 | 26.56 | 30.35 | 49.22 | 57.33 | 25.91 | 33.12 | - |
| ◉ ⊞ R1-Searcher | 27.34 | 33.96 | 39.84 | 46.36 | 10.16 | 16.63 | 32.03 | 44.93 | 41.41 | 47.12 | 56.25 | 64.76 | 34.51 | 42.29 | 51.26 |
| ◉ ⊞ Search-R1 | 35.15 | 38.21 | 43.77 | 51.26 | 17.18 | 21.45 | **38.34** | 43.79 | 43.75 | 47.03 | 51.56 | 61.03 | 38.29 | 43.80 | 53.06 |
| ◉ ⊞ + SIT | 56.25 | 60.75 | 54.68 | 60.44 | 32.81 | 41.54 | 34.37 | 48.97 | 46.87 | 51.17 | 62.50 | 69.79 | 47.91 | 55.44 | **65.02** |
| ◉ ⚒ Δ | +21.10 | +22.54 | +10.91 | +9.18 | +15.63 | +20.09 | -3.97 | +5.18 | +3.12 | +4.14 | +10.94 | +8.76 | +9.62 | +11.64 | +11.96 |
| ◉ ⊞ Search-R1-PPO | 39.84 | 42.38 | 47.66 | 56.28 | 21.09 | 32.91 | 18.75 | 32.27 | 39.08 | 44.26 | 60.15 | 69.29 | 37.76 | 46.23 | 49.31 |
| ◉ ⊞ + SIT | 57.03 | 61.47 | 52.34 | 57.83 | 30.47 | 35.32 | 33.59 | 46.84 | **49.22** | 52.34 | 61.71 | 69.62 | 47.39 | 53.90 | **65.02** |
| ◉ ⚒ Δ | +17.19 | +19.09 | +4.68 | +1.55 | +9.38 | +2.41 | +14.84 | +14.57 | +10.14 | +8.08 | +1.56 | +0.33 | +9.63 | +7.67 | +15.71 |
| ◉ ⚒ Graph-R1 | 55.47 | 65.04 | 57.03 | 62.69 | 36.72 | 46.17 | 33.59 | 49.87 | 45.31 | 51.22 | 63.28 | 71.93 | 48.57 | 57.82 | 60.40 |
| ◉ ⚒ + SIT | 60.94 | 68.26 | 59.38 | 66.14 | 40.63 | 51.63 | 38.28 | **51.99** | 49.21 | 53.49 | 64.06 | 72.37 | 52.08 | 60.65 | 64.90 |
| ◉ ⚒ Δ | +5.47 | +3.22 | +2.35 | +3.45 | +3.91 | +5.46 | +4.69 | +2.12 | +3.90 | +2.27 | +0.78 | +0.44 | +3.51 | +2.83 | +4.50 |
| *Qwen2.5-14B-Instruct* | | | | | | | | | | | | | | | |
| ◉ ⚒ Graph-R1 | 67.97 | 75.46 | 67.19 | 72.52 | 43.75 | 57.54 | 39.84 | 53.81 | 49.22 | 53.33 | 68.75 | 76.43 | 56.12 | 64.85 | 60.65 |
| ◉ ⚒ + SIT | **70.31** | **77.12** | **68.75** | **74.47** | **45.31** | **57.88** | **40.63** | **56.02** | **50.00** | **54.06** | **71.09** | **77.84** | **57.68** | **66.23** | **65.13** |
| ◉ ⚒ Δ | +2.34 | +1.66 | +1.56 | +1.95 | +1.56 | +0.34 | +0.79 | +2.21 | +0.78 | +0.73 | +2.34 | +1.41 | +1.56 | +1.38 | +4.48 |

used in the Graph-R1 setting. We also run SIT on Search-R1 in the Graph-R1 setting with a smaller, dataset-specific chunk-based retreival set.

## 5.1 IMPLEMENTATIONS

**Baselines.** In Graph-R1 setting, we follow the previous work, including training-free methods from Graph-R1: NaiveGeneration, StandardRAG(Lewis et al., 2020), GraphRAG(Edge et al., 2025), LightRAG(Guo et al., 2025), PathRAG(Chen et al., 2025), HippoRAG2(Gutiérrez et al., 2025), HyperGraphRAG(Luo et al., 2025b) , training:SFT(Zheng et al., 2024), R1(Shao et al., 2024a), R1-Searcher(Song et al., 2025) and Graph-R1(Luo et al., 2025a) itself, we cite their performances for comparison if not specified. In the Search-R1 setting, additional baselines including CoT(Wei et al., 2022), IRCoT(Trivedi et al., 2022b), Search-o1(Li et al., 2025a), and Rejection Sampling(Ahn et al., 2024) is compared. Detailed description of these baselines are put in the Appendix E. We use Qwen2.5-7B-Instruct(Qwen et al., 2025) and Qwen2.5-14B-Instruct as LLM backbone for training. We also have done additional experiments on Qwen3(Yang et al., 2025) in Appendix C.1 and Section 5.4.

**Retriever.** The retriever we used is highly dependent on the backbone. In Search-R1, the retriever is E5(Wang et al., 2022). In Graph-R1, the retriever is hypergraph-based retrieval with bge-large-en-v1.5(Chen et al., 2023).

**Datasets and Metrics.** Due to the different dataset splitting protocols in Search-R1 and Graph-R1, we conduct our experiments under both settings to ensure fair comparison. In Graph-R1 setting, we follow the original paper setting and use 6 common datasets(Jin et al., 2025b) for QA, including 2Wikihop(Ho et al., 2020), HotpotQA(Yang et al., 2018), Musique(Trivedi et al., 2022a), NQ(Kwiatkowski et al., 2019), PopQA(Mallen et al., 2023), TriviaQA(Joshi et al., 2017). Also in this setting we compare with Search-R1 baselines. We use EM, F1 and R-S to evaluate results. EM and F1 measures the answer and R-S measures the retrieval performances. In Search-R1 setting, we follow the original paper setting, appending one new dataset Bamboogle(Press et al., 2022), and using F1 score for comparison. Detailed information are referred to Appendix E.

## 5.2 COMPARISON IN GRAPH-R1 SETTING

We show the results in Table 2. Note that Search-R1 uses PPO method in its paper but Graph-R1 runs GRPO in their experiments so we run the Search-R1-PPO by ourselves as the PPO variants in the table. We found that SIT improves the performance of Graph-R1 by an average of 3 F1 points, Search-R1 by an average of 11 F1 points and Search-R1-PPO by an average of 7 F1 points, demonstrating a substantial performance gain across different RL methods. Also, the improvement in R-S scores indicates that SIT can actually improve the exploitation in focusing retrieval necessary information.

Then we analysis the R-S of SIT compared with Graph-R1 in Table 3. This suggests that SIT's performance gains are partially driven by improved retrieval quality.

Table 3: R-S comparison of SIT.

|  | **2Wiki** | **HotpotQA** | **Musique** | **NQ** | **PopQA** | **TriviaQA** | **Avg.** |
|---|---|---|---|---|---|---|---|
| Graph-R1 | 55.24 | 56.27 | 52.95 | 69.25 | 61.55 | 67.16 | 60.40 |
| +SIT | **60.69** | **60.36** | **61.54** | **72.86** | **64.97** | **68.99** | **64.90** |

## 5.3 COMPARISON IN SEARCH-R1 SETTING

Table 4: Main results (F1 scores) compared in Search-R1 setting. The best performance is set in bold. $^{\dagger}/^{\star}$ represents IND/OOD datasets. Icons have the same meaning as Table 2.

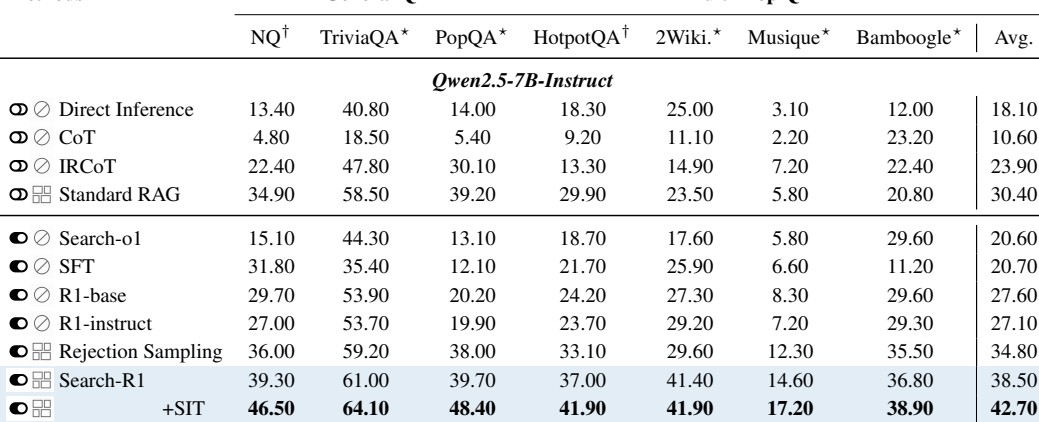

| Methods | General QA | | | | Multi-Hop QA | | | |
|---|---|---|---|---|---|---|---|---|
|  | NQ$^{\dagger}$ | TriviaQA$^{\star}$ | PopQA$^{\star}$ | HotpotQA$^{\dagger}$ | 2Wiki.$^{\star}$ | Musique$^{\star}$ | Bamboogle$^{\star}$ | Avg. |
| *Qwen2.5-7B-Instruct* | | | | | | | | |
| Direct Inference | 13.40 | 40.80 | 14.00 | 18.30 | 25.00 | 3.10 | 12.00 | 18.10 |
| CoT | 4.80 | 18.50 | 5.40 | 9.20 | 11.10 | 2.20 | 23.20 | 10.60 |
| IRCoT | 22.40 | 47.80 | 30.10 | 13.30 | 14.90 | 7.20 | 22.40 | 23.90 |
| Standard RAG | 34.90 | 58.50 | 39.20 | 29.90 | 23.50 | 5.80 | 20.80 | 30.40 |
| Search-o1 | 15.10 | 44.30 | 13.10 | 18.70 | 17.60 | 5.80 | 29.60 | 20.60 |
| SFT | 31.80 | 35.40 | 12.10 | 21.70 | 25.90 | 6.60 | 11.20 | 20.70 |
| R1-base | 29.70 | 53.90 | 20.20 | 24.20 | 27.30 | 8.30 | 29.60 | 27.60 |
| R1-instruct | 27.00 | 53.70 | 19.90 | 23.70 | 29.20 | 7.20 | 29.30 | 27.10 |
| Rejection Sampling | 36.00 | 59.20 | 38.00 | 33.10 | 29.60 | 12.30 | 35.50 | 34.80 |
| Search-R1 | 39.30 | 61.00 | 39.70 | 37.00 | 41.40 | 14.60 | 36.80 | 38.50 |
| +SIT | **46.50** | **64.10** | **48.40** | **41.90** | **41.90** | **17.20** | **38.90** | **42.70** |
| Δ | *+7.20* | *+3.10* | *+8.70* | *+4.90* | *+0.50* | *+2.60* | *+2.10* | *+4.20* |

In Search-R1 setting, we show the results of using Fullwiki as the retrieval set to show our methods' robustness in retrieval set. As constructing a full Wikipedia hypergraph in the manner of Graph-R1 is currently computationally prohibitive, we only use Search-R1 as our backbone. The results shows that SIT also can increase performances when the retrieval set is very large, and can show incremental performances in both IND and OOD datasets in Table 4. Notably, SIT improves the performance of Search-R1 by an average of 4 F1 points and is achieved within only 300 training steps, whereas the original Search-R1 requires 1000 steps, highlighting the improved efficiency of our approach.

## 5.4 TRAINING-FREE SIT

To demonstrate versatility and scalability, we evaluate SIT as a *training-free* inference strategy on larger models where RL fine-tuning is computationally prohibitive. By enforcing an initial retrieval step before reasoning, SIT consistently delivers substantial gains across benchmarks (Table 5). These results confirm that "plan failure" from ungrounded thinking persists even in large-scale models, and SIT serves as a robust, plug-and-play solution to mitigate hallucinations and enhance reasoning stability without parameter updates.

Table 5: Performance (F1 Score) of SIT as a training-free inference strategy on large-scale models. SIT consistently improves performance across all datasets without any parameter updates.

| Model | 2Wiki | HotpotQA | Musique | NQ | PopQA | TriviaQA |
|---|---|---|---|---|---|---|
| Qwen2.5-32B-Instruct | 13.73 | 23.96 | 8.29 | 11.62 | 15.19 | 23.65 |
| + SIT | **18.17** | **26.14** | **13.04** | **15.63** | **17.08** | **27.84** |
| Qwen3-235-A30B-Instruct | 30.56 | 37.80 | 19.93 | 21.49 | 28.73 | 38.55 |
| + SIT | **38.39** | **48.82** | **28.17** | **24.68** | **33.61** | **44.72** |

## 6 ABLATIONS

Experiments are done in the Graph-R1 setting in the ablation section, and we aim to answer the following three questions:

- Q1. Why Search-Initialized Thinking can make the performance better, from an entropy perspective.

- Q2. Can Initial Knowledge shorten the number of thinking turns? And what is metrics' dynamics in every step in the training?

- Q3. Will Search-Initialized Thinking in RL training downgrade the generalization of trained models?

### 6.1 ENTROPY ANALYSIS

In RL training, the entropy demonstrates model's exploration ability in training. However, in the context of multi-hop RAG, unconstrained exploration is not always beneficial, as the reasoning process must remain aligned with the information available in the retrieval set. SIT is designed precisely to provide this initial alignment. We show the comparison of Graph-R1's entropy of tokens between "<answer>...</answer>", "<think>...</think>", "<query>...</query>" with SIT or not in Figure 3.

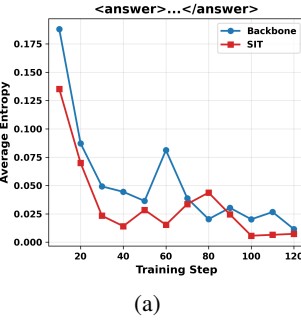 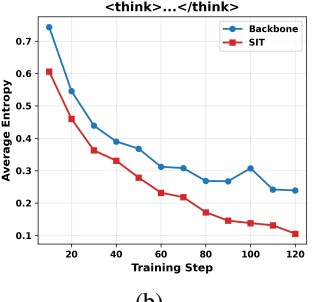 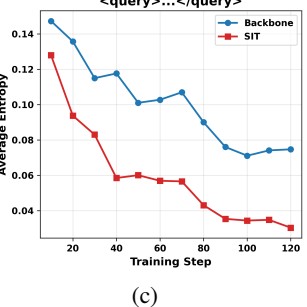

(a)  (b)  (c)

Figure 3: Entropy comparison of backbone (Graph-R1) and SIT. (a), (b), and (c) show average entropy of tokens between "<answer>...</answer>", "<think>...</think>", "<query>...</query>".

We found that the entropy values for all action types are generally lower with SIT than without it. At zero step with the same LLM, the lower entropy of tokens between "<answer>" "</answer>" (which is actually the answer tokens) of SIT fits the intermediate conclusion in the proof in Appendix D that

$$\mathbb{E}_\pi \left[ I(A^\star; \mathcal{H}_T^{SIT} \mid Q) \right] \geq \mathbb{E}_\pi \left[ I(A^\star; \mathcal{H}_T \mid Q) \right], \tag{5}$$

which predicts the lower entropy of SIT answer tokens. Although there is a single training step where the answer entropy for SIT is momentarily higher, the overarching trend shows that SIT consistently leads to lower answer token entropy.

Besides, the lower entropy of think and search tokens show that LLM with SIT has more determined exploration direction in thinking and searching, which is exactly what we assume in the beginning.

## 6.2 SHORTER TURNS AND METRICS DYNAMICS.

We show that with SIT, the exploration turns of LLMs shrinks about one turn on average in Table 6. Shorter turns means less noise in the retrieval that can make LLM more focus on the right information. Next, we show the F1 and R-S scores in the training step in Figure 4. We found that with SIT, our

Table 6: Average turns of Graph-R1 with or without SIT.

|  | 2Wiki | HotpotQA | Musique | NQ | PopQA | TriviaQA | Avg. |
|---|---|---|---|---|---|---|---|
| Graph-R1 | 3.12 | 3.12 | 3.88 | 3.06 | 3.53 | 2.82 | 3.26 |
| +SIT | **2.72** | **2.80** | **2.68** | **1.52** | **1.91** | **1.72** | **2.22** |

model's RS is high from the beginning. Even when we exclude the initial knowledge in computing the metrics, the R-S score of backbone with SIT can still increase to a higher value than the model without SIT.

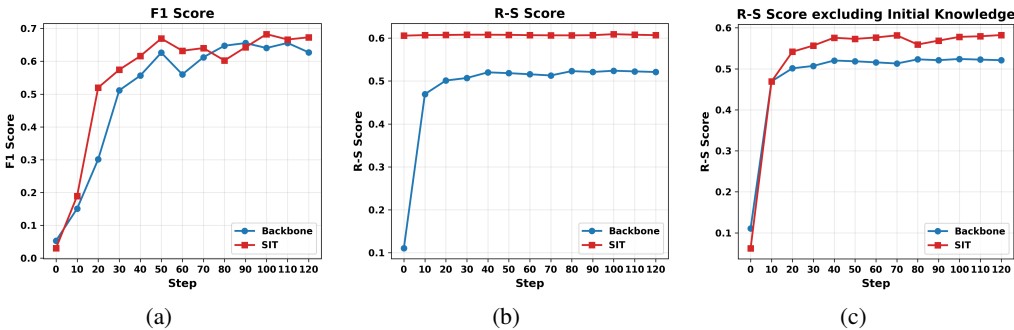

(a)                              (b)                              (c)

Figure 4: F1 and R-S scores per training step on the 2Wiki dataset. (a) F1 score. (b) R-S score. (c) R-S score excluding the initial knowledge.

## 6.3 GENERALIZATION

### 6.3.1 GENERALIZATION ACROSS DATASETS

While the generalization performance on OOD datasets using the Search-R1 backbone was presented in Table 4, this section evaluates the generalization of SIT with the Graph-R1 backbone. The results show that our method not only achieves better results in IID conditions but also show better generalization results on average than without SIT.

### 6.3.2 MISMATCHED INITIAL KNOWLEDGE

We further investigate the robustness of Search-Initialized Thinking (SIT) against variations in the quality and source of the initial knowledge $P_0$.

**Noisy IK.** In real-world scenarios, the initial retrieval $P_0$ may contain irrelevant information or noise. To simulate this, we conduct experiments using the full Wikipedia corpus as the retrieval source for the initial step (denoted as SIT-wiki), which introduces significantly more noise compared to the dataset-specific retrieval sets. As shown in Table 8, although the introduction of noise in SIT-wiki leads to a slight performance drop compared to the standard SIT, it still consistently outperforms the baseline without SIT across all datasets. This demonstrates that the benefit of SIT comes from the *grounding* effect of the initial thought, which remains effective even when the initial search is imperfect.

**Mismatched Retriever.** To verify that our improvements are not dependent on a specific retrieval model, we evaluate SIT using different dense retrievers. We compare the default BGE retriever (SIT-bge) with the E5 retriever (SIT-e5). Table 9 presents the results across six datasets. We observe that SIT yields consistent performance gains regardless of the retriever used, confirming that the SIT framework is retriever-agnostic and generalizes well across different semantic embedding spaces.

Table 7: Generalization test on backbone and SIT. The row datasets are training datasets and the column datasets are test datasets.

| Train Datasets | 2Wiki. | HotpotQA | Musique | NQ | PopQA | TriviaQA | Avg. |
|---|---|---|---|---|---|---|---|
| 2Wiki. | 65.04 | 59.92 | 35.92 | 45.24 | 42.57 | 63.38 | 52.01 |
| +SIT | **68.26** | 63.90 | 44.53 | 46.89 | 50.78 | 65.53 | **56.65** |
| +Δ | *+3.22* | *+3.98* | *+8.61* | *+1.65* | *+8.21* | *+2.15* | *+4.64* |
| HotpotQA | 58.27 | 62.69 | 33.27 | 37.89 | 44.30 | 57.20 | 48.94 |
| +SIT | 60.86 | **66.14** | 38.87 | 45.14 | 47.60 | 66.96 | 54.26 |
| +Δ | *+2.59* | *+3.45* | *+5.60* | *+7.25* | *+3.30* | *+9.76* | *+5.32* |
| Musique | 43.87 | 52.32 | 46.17 | 43.66 | 44.76 | 64.45 | 49.21 |
| +SIT | 54.90 | 59.99 | **51.63** | 47.63 | 48.98 | 69.82 | 55.49 |
| +Δ | *+11.03* | *+7.67* | *+5.46* | *+3.97* | *+4.22* | *+5.37* | *+6.28* |
| NQ | 52.13 | 53.19 | 34.57 | 49.87 | 43.10 | 63.74 | 49.43 |
| +SIT | 54.77 | 55.83 | 37.75 | **51.99** | 48.72 | 67.38 | 52.74 |
| +Δ | *+2.64* | *+2.64* | *+3.18* | *+2.12* | *+5.62* | *+3.64* | *+3.31* |
| PopQA | 47.41 | 58.45 | 35.99 | 43.40 | 51.22 | 68.91 | 50.90 |
| +SIT | 48.51 | 57.52 | 34.66 | 43.88 | **53.49** | 69.98 | 51.34 |
| +Δ | *+1.10* | *-0.93* | *-1.33* | *+0.48* | *+2.27* | *+1.07* | *+0.44* |
| TriviaQA | 46.83 | 53.82 | 22.87 | 41.66 | 44.71 | 71.93 | 46.97 |
| +SIT | 52.17 | 55.18 | 31.31 | 44.87 | 47.23 | **72.37** | 50.52 |
| +Δ | *+5.34* | *+1.36* | *+8.44* | *+3.21* | *+2.52* | *+0.44* | *+3.55* |

Table 8: Performance(F1 Score) comparison with noisy initial knowledge.

| Method | 2Wiki | HotpotQA | Musique | NQ | PopQA | TriviaQA |
|---|---|---|---|---|---|---|
| Qwen2.5-7B-Instruct | 65.04 | 62.69 | 46.17 | 49.87 | 51.22 | 71.93 |
| + SIT (Standard) | **68.26** | **66.14** | **51.63** | **51.99** | 53.49 | **72.37** |
| + SIT-wiki (Noisy) | 66.18 | 62.91 | 47.16 | 50.43 | **53.98** | 71.77 |

Table 9: Ablation study on retriever quality.

| Method | 2Wiki | HotpotQA | Musique | NQ | PopQA | TriviaQA |
|---|---|---|---|---|---|---|
| Qwen2.5-7B-Instruct | 65.04 | 62.69 | 46.17 | 49.87 | 51.22 | 71.93 |
| SIT-bge (Standard) | **68.26** | **66.14** | 51.63 | **51.99** | **53.49** | **72.37** |
| SIT-e5 | 68.18 | 64.74 | **54.27** | 50.74 | 53.46 | 72.21 |

# 7 CONCLUSION

All in all, we propose an easy but effective module in iterative RAG pipeline called Search-Initialized Thinking (SIT) that can guide right directions of thinking, resulting in more efficient exploration in RL training and better end-to-end performances. Our comprehensive experiments rigorously validate the efficacy and robustness of SIT. The approach delivers substantial performance gains to state-of-the-art RL-based frameworks, including Search-R1 and Graph-R1, across diverse RL algorithms (PPO and GRPO) and varied retrieval contexts—from small, structured corpora to large-scale, unstructured document sets. A key finding is that this performance enhancement does not come at the cost of generalization; in fact, SIT consistently maintains or even improves upon the generalization capabilities of the backbone models, showcasing its reliability. Crucially, we also demonstrate SIT's scalability as a plug-and-play, training-free strategy for large models. This motivates us the shift of designing advanced RAG systems: from a "think-then-search" model to a more synergistic "search-informed thinking" process.

## 8 ETHICS STATEMENT

Regarding broader societal impact, while improved RAG systems can increase the reliability and trustworthiness of AI-generated information, there is of course some risk that they may also enable more convincing misinformation if misused.

## 9 REPRODUCIBILITY STATEMENT

We present a detailed training algorithm in Appendix B, technical proofs in Appendix D, and additional experimental/implementation details in Appendix E. Additionally, code for our model is uploaded as supplemental materials with the submission.

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

## A LLM USAGE

Large Language Models (LLMs) were used to aid in the writing and polishing of the manuscript. Specifically, we used an LLM to assist in refining the language, improving readability, and ensuring clarity in various sections of the paper. The model helped with tasks such as sentence rephrasing, grammar checking, and enhancing the overall flow of the text.

It is important to note that the LLM was not involved in the ideation, research methodology, or experimental design. All research concepts, ideas, and analyses were developed and conducted by the authors. The contributions of the LLM were solely focused on improving the linguistic quality of the paper, with no involvement in the scientific content or data analysis.

The authors take full responsibility for the content of the manuscript, including any text generated or polished by the LLM. We have ensured that the LLM-generated text adheres to ethical guidelines and does not contribute to plagiarism or scientific misconduct.

---

**Algorithm 1** Search-Initialized Thinking

---

**Require:** Input $x$, LLM $\pi_\theta$, Retrieval set $\mathcal{R}$, Max turns $B$.
**Ensure:** Output $y$.
 1: Initialize $y \leftarrow \emptyset$
 2: Initialize $b \leftarrow 0$
 3: Initialize Searching Knowledge $\mathcal{P}_0 = \mathcal{R}(x)$ and update $x \leftarrow x + \mathcal{P}_0$
 4: **while** $b < B$ **do**
 5:     Rollout $y_b \leftarrow \emptyset$
 6:     **while** True **do**
 7:         Generating $y_t \sim \pi_\theta(\cdot \mid x, y + y_b)$
 8:         concatenate token $y_b \leftarrow y_b + y_t$
 9:         **if** $y_t$ in [`</query>`, `</answer>`, `<eos>`] **then** break
10:         **end if**
11:     **end while**
12:     $y \leftarrow y + y_b$
13:     **if** extract `<query> </query>` from $y_b$ **then**
14:         Extract $q \leftarrow \text{Parse}(y_b, $`<query>`$, $`</query>`$)$
15:         Retrive knowledge $d = \mathcal{R}(q)$
16:         Continue rollout $y \leftarrow y + $`</knowledge>`$d$`</knowledge>`
17:     **else if** extract `</answer>` from $y_b$ **then**
18:         **return** $y$
19:     **end if**
20:     count turns $b \leftarrow b + 1$
21: **end while**
22: **return** $y$

---

# B ALGORITHM

# C ADDITIONAL EXPERIMENTS

## C.1 QWEN3 MODEL RESULTS

We show the Qwen3-4B-Instruct-2507 model's performances in the training step in Figure C.1. It is shown that even bad results, SIT can still improve Qwen3 performances. We check the output of Qwen3 and find that the reason is that Qwen3 instruction models have used "think" token in its pre-train so when they have removed think pattern in 2507 model, it's hard for the model to generate the thinking process in the pipeline, resulting in low performances.

## C.2 CASE STUDY

In this section, we show a classical example of why search-initialized thinking is useful. In Graph-R1, when the model lacks planning ability to split the question into two parts, it will generate a useless searching for both two things in turns and turns. As shown in Table 10, it fails to retrieve the directors. While as shown in Table 11 the model with SIT knows searching for two things is useless, then it will split the question and search for two directors separately and finally retrieve the right documents, resulting in the right answer.

# D THEORETICAL PROOF

**Proposition 1.** Search-Initialized Thinking is better than Model-Initialized thinking in iterative RAG from an entropy perspective.

*Proof.* Given the condition of iterative RAG for an LLM $\pi$ divides the budget across $T$ rounds as $B = \sum_{t=1}^{T} B_t$. At each round $t >= 1$, we denote $\mathcal{P}_t$ as the retrieval results at this step, and the prior evidence $\mathcal{H}_{t-1} = \{\mathcal{P}_1, \ldots, \mathcal{P}_{t-1}\}$. The LLM uses $\mathcal{H}_{t-1}$ to update its internal belief $h_{t-1}$ and

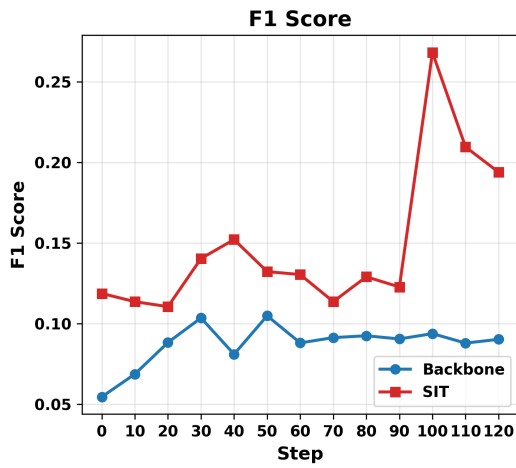

Figure 5: Qwen3-4B-Instruct-2507 model's F1 score in each step in 2Wiki dataset. Backbone is Graph-R1.

Table 10: A case study of Graph-R1.

| |
|---|
| **Question**: Which film has the director born later, I'Ll Tell The World or Saranggola? |
| **Ground Truth**: "Saranggola" |
| **Graph-R1**: |
| `<think>` To answer this question, I need to find information about the birth years of the directors of the movies I'Ll Tell The World and Saranggola, then compare them. `</think>` |
| `<query>` SELECT directorBirthYear WHERE movieName = "I'Ll Tell The World" OR movieName = "Saranggola" `</query>` |
| `<knowledge>` Invalid tool call format. Please use `<query>`{ "query": "statement" }`</query>` format. `</knowledge>` |
| `<think>` The previous query did not work as expected, so I'll try a different approach. First, I need to get the birth years of the directors of both movies. `</think>` |
| `<query>` SELECT directorBirthYear WHERE movieName = "I'Ll Tell The World" OR movieName = "Saranggola" `</query>` |
| `<knowledge>` Invalid tool call format. Please use `<query>`{ "query": "statement" }`</query>` format. `</knowledge>` |
| `<think>` Given the error in my previous attempts, I'll directly query for the birth years of the directors of I'Ll Tell The World and Saranggola, and then compare them. `</think>` |
| `<query>` { "query": "SELECT directorBirthYear WHERE movieName = 'I'Ll Tell The World' OR movieName = 'Saranggola'" } `</query>` |
| `<knowledge>` { "results": [ { "`<knowledge>`": "Lena Dunham (born May 13, 1986) is an American actress, writer, director, and producer.", "`<coherence>`": 1.0 }, { "`<knowledge>`": "Priyadarshan (born January 30, 1957) is an Indian film director, producer, and screenwriter.", "`<coherence>`": 0.508 }, { "`<knowledge>`": "V. S. Raghavan (director)", "`<coherence>`": 0.333 } ] } `</knowledge>` |
| `<think>` I found the birth years of the directors of both movies. Lena Dunham is the director of I'Ll Tell The World and was born in 1986. While V. S. Raghavan is the director of Saranggola and was born in 1935. Since Lena Dunham's birth year is later than V. S. Raghavan, the film I'Ll Tell The World has the director born later. `</think>` |
| `<answer>` I'Ll Tell The World `</answer>` |

selects new evidence $\mathcal{P}_t$ of size $B_t$ by actively exploring the graph based on current uncertainty. The updated belief $h_t$ is obtained via Bayesian inference, and the entire process forms a dynamic system:

$$h_t = f(h_{t-1}, \mathcal{P}_t, R_G). \tag{6}$$

To evaluate retrieval progress, we define a Lyapunov-style potential function $V_t = H(A^\star \mid Q, \mathcal{H}_t)$, which quantifies the remaining uncertainty after round $t$. Each retrieval step reduces entropy by:

$$V_{t-1} - V_t = I(A^\star; \mathcal{P}_t \mid Q, \mathcal{H}_{t-1}), \tag{7}$$

We focus on the first step of iterative RAG that $t = 1$. The entropy reduction for the first step is

$$V_0 - V_1 = I(A^\star; \mathcal{P}_1 \mid Q, \mathcal{H}_0). \tag{8}$$

Table 11: A case study of Graph-R1+SIT.

**Question**: Which film has the director born later, I'll Tell The World or Saranggola?

**Ground Truth**: "Saranggola"

**Graph-R1+SIT**:

`<think>` From the knowledge provided, I'll Tell The World is a 1945 American comedy film directed by Leslie Goodwins. Saranggola is a 1999 Filipino drama film directed by Gil Portes. To determine which film has the director born later, I will need to find the birth years of Leslie Goodwins and Gil Portes. `</think>`

`<query>` { "type": "search", "query": "Leslie Goodwins birth year" } `</query>`

`<knowledge>` {"results": [{""<knowledge>"": "They Met in Argentina is a 1941 American film directed by Leslie Goodwins and Jack Hively for RKO Pictures.", "<coherence>": 1.7}, {"<knowledge>": "Leslie Goodwins", "<coherence>": 1.0}, {"<knowledge>": "Leslie Goodwins (17 September 1899 – 8 January 1969) was an English film director and screenwriter.", "<coherence>": 0.819}, {"<knowledge>": "With Love and Kisses is a 1936 American musical comedy film directed by Leslie Goodwins and starring Pinky Tomlin, Toby Wing and Kane Richmond.", "<coherence>": 0.643}, {"<knowledge>": "I'll Tell the World is a 1945 American comedy film directed by Leslie Goodwins and written by Henry Blankfort and Lester Pine.", "<coherence>": 0.333}, {"<knowledge>": "The Mexican Spitfire's Baby is a 1941 American comedy film directed by Leslie Goodwins.", "<coherence>": 0.25}, {"<knowledge>": "Leslie Goodwins is acknowledged but lacks detailed background in the provided text.", "<coherence>": 0.2}, {"<knowledge>": "Murder in the Blue Room is a 1944 American film directed by Leslie Goodwins. It is the second remake of the 1933 American Pre-Code murder-mystery film 'Secret of the Blue Room.'", "<coherence>": 0.167}]} `</knowledge>`

`<think>` From the knowledge provided, Leslie Goodwins was born on September 17, 1899. Now I need to find the birth year of Gil Portes to determine which film has the director born later. `</think>`

`<query>` {"type": "search", "query": "Gil Portes birth year"} `</query>`

`<knowledge>` {"results": [{"<knowledge>": "Gil M. Portes (September 13, 1945 – May 24, 2017) was a Filipino film director, film producer and screenwriter.", "<coherence>": 1.125}, {"<knowledge>": "Saturday Night (1950 film) is a 1950 Spanish drama film directed by Rafael Gil.", "<coherence>": 1.0}, {"<knowledge>": "The Man Who Wanted to Kill Himself (1970 film)", "<coherence>": 0.5}, {"<knowledge>": "Carlos Gardel (born Charles Romuald Gardès; 11 December 1890 – 24 June 1935) was a French Argentine singer, songwriter, composer and actor.", "<coherence>": 0.5}, {"<knowledge>": "Rafael Gil was a Spanish film director and screenwriter.", "<coherence>": 0.4}, {"<knowledge>": "The Autonomines is a Spanish comedy film directed by Rafael Gil and starring Alfredo Landa, María Casanova, and Manolo Codeso.", "<coherence>": 0.333}, {"<knowledge>": "Gil was a prominent director of the Franco era.", "<coherence>": 0.333}, {"<knowledge>": "El canto del gallo is a 1955 Spanish drama film directed by Rafael Gil.", "<coherence>": 0.25}, {"<knowledge>": "Gil's film 'La noche del sábado' was nominated for the Gold Lion at the 1950 Venice Film Festival.", "<coherence>": 0.25}, {"<knowledge>": "Saranggola (international title: The Kite) is a 1999 Filipino drama film directed by Gil Portes, starring Ricky Davao, Lester Llansang and Jennifer Sevilla.", "<coherence>": 0.167}]} `</knowledge>`

`<think>` From the knowledge provided, Leslie Goodwins was born on September 17, 1899. Gil Portes was born on September 13, 1945. Now I can compare the birth years to determine which film has the director born later. `</think>`

`<answer>` Saranggola `</answer>`

In Model-Initialized thinking, $\mathcal{H}_0 = \{\emptyset\}$ while in our Search-Initialized Thinking, the $\mathcal{H}_0 = \{\mathcal{P}_0\}$. Thus

Summing over all rounds, the total information gain of the adaptive strategy satisfies:

$$\mathbb{E}_\pi \left[ I(A^\star; \mathcal{H}_T^{SIT} \mid Q) \right] = \mathbb{E}_\pi \left[ \sum_{t=1}^{T} I(A^\star; \mathcal{P}_t^{SIT} \mid Q, \mathcal{H}_{t-1}^{SIT}) \right] \tag{9}$$

$$\geq \mathbb{E}_\pi \left[ \sum_{t=1}^{T} I(A^\star; \mathcal{P}_t \mid Q, \mathcal{H}_{t-1}) \right] \tag{10}$$

$$= \mathbb{E}_\pi \left[ I(A^\star; \mathcal{H}_T \mid Q) \right], \tag{11}$$

while the unequality comes from the fact that with $\mathcal{H}_0 = \{\mathcal{P}_0\}$, which is highly related to $Q$, at each step $t >= 1$,

$$I(A^\star; \mathcal{P}_t^{SIT} \mid Q, \mathcal{H}_{t-1}^{SIT}) \geq I(A^\star; \mathcal{P}_t \mid Q, \mathcal{H}_{t-1}), \tag{12}$$

which means the SIT is no worse than the MIT.

Let $\rho_T$ denote the information gain per token at the end of the iterative operation:

$$\rho_T = \frac{I(A^\star; \mathcal{H}_T \mid Q)}{B}, \tag{13}$$

From a Bayesian viewpoint, retrieval efficiency can be seen as how much uncertainty is reduced per token. SIT achieves a greater entropy reduction under the same budget, or requires fewer tokens to reach the same posterior certainty, it is strictly more efficient. Moreover, by Fano's inequality,

$$P_e \leq \frac{H(A^\star \mid Q) - I(A^\star; \mathcal{H}_T \mid Q) + 1}{\log |\mathcal{A}|}, \tag{14}$$

we conclude that the lower the conditional entropy, the lower the expected error. Therefore, greater mutual information directly translates into improved answer accuracy.

In conclusion, Search-Initialized Thinking enables the agent to get more information gain and lower entropy at the end of iterative RAG, leading to more efficient and accurate question answering. □

## E    DETAILED IMPLEMENTATIONS AND HYPERPARAMETERS

### E.1    BASELINES IN GRAPH-R1 SETTING

Baselines in Graph-R1 setting first utilizes **GPT-4o-mini** as the inference-only generator. This includes **NaiveGeneration**, which performs zero-shot generation without retrieval to evaluate the base model's capacity, and **StandardRAG** (Lewis et al., 2020), a conventional chunk-based retrieval-augmented generation approach. We also include several graph-based retrieval methods: **GraphRAG** (Edge et al., 2025), which constructs entity graphs for one-shot retrieval; **LightRAG** (Guo et al., 2025), a lightweight variant that builds compact graphs for more efficient retrieval; **PathRAG** (Chen et al., 2025), which performs retrieval via path-based pruning on entity graphs; **HippoRAG2** (Gutiérrez et al., 2025), which employs a hierarchical path planner over knowledge graphs to improve retrieval efficiency; and **HyperGraphRAG** (Luo et al., 2025b), which constructs n-ary relational hypergraphs to support a single retrieval step.

The second set of baselines is based on the `Qwen2.5-Instruct (7B)` model. We begin with foundational methods, including a **NaiveGeneration** approach as a lower-bound, the classic **StandardRAG** (Lewis et al., 2020) pipeline, and **SFT** (Zheng et al., 2024), which involves supervised fine-tuning on QA pairs. Furthermore, we evaluate several advanced methods trained with reinforcement learning: **R1** (Shao et al., 2024a), a GRPO-trained policy that generates answers directly without retrieval; **Search-R1** (Jin et al., 2025a), a multi-turn chunk-based retrieval method trained with GRPO; **R1-Searcher** (Song et al., 2025), a two-stage GRPO-based method for chunk-based retrieval; and **Graph-R1**(Luo et al., 2025a), an agentic GraphRAG framework enhanced by end-to-end reinforcement learning.

### E.2    BASELINES IN SEARCH-R1 SETTING

In Search-R1 setting, despite the baselines in last section, we also compare against prominent reasoning and generation strategies: **CoT** (Wei et al., 2022): reasoning with chain of thought; **IRCoT**(Trivedi et al., 2022b): reasoning with chain of thought with retreival; **Search-o1**(Li et al., 2025a): integrating an agentic search workflow into the reasoning process; and **Rejection Sampling**(Ahn et al., 2024): SFT on trajectories that succeed.

### E.3    METRICS

**Exact Match (EM).** This metric provides a strict evaluation of answer accuracy. It determines if the generated answer $y_i$ is identical to the ground-truth reference $y_i^\star$ after both have undergone a normalization process. This process typically includes lowercasing, removing punctuation, and standardizing whitespace. The score is 1 if they match perfectly, and 0 otherwise. The final EM score is the average over all samples:

$$\text{EM} = \frac{1}{N} \sum_{i=1}^{N} \mathbb{I} \left\{ \text{norm}(y_i) = \text{norm}(y_i^\star) \right\}. \tag{15}$$

**F1 Score.** Unlike the all-or-nothing EM, the F1 score offers a more nuanced measure of quality by assessing the word-level (token) overlap between the prediction and the ground truth. It calculates the harmonic mean of precision (the fraction of predicted tokens that are correct) and recall (the fraction of ground-truth tokens that are predicted), providing a balanced assessment of token accuracy:

$$\text{F1} = \frac{1}{N} \sum_{i=1}^{N} \frac{2 \cdot |\text{tokens}(y_i) \cap \text{tokens}(y_i^\star)|}{|\text{tokens}(y_i)| + |\text{tokens}(y_i^\star)|}. \tag{16}$$

**Retrieval Similarity (R-S).** This metric evaluates the quality of the retrieval component of the RAG system, rather than the final generated answer. It measures the semantic relevance of the retrieved context $k_{\text{retr}}^{(i)}$ compared to the ideal "gold" context $k_{\text{gold}}^{(i)}$. To do this, both texts are converted into vector representations using a semantic embedding function $\text{Enc}(\cdot)$, and their cosine similarity is computed:

$$\text{R-S} = \frac{1}{N} \sum_{i=1}^{N} \cos\left(\text{Enc}(k_{\text{retr}}^{(i)}), \text{Enc}(k_{\text{gold}}^{(i)})\right). \tag{17}$$

### E.4 HYPERPARAMETERS

We show in Table 12 the hyperparameters in Graph-R1 setting. In Search-R1 setting, the hyperparameters are the same as Search-R1. The models with SIT share the same hyperparameters with the backbone method.

| Method | Backbone | Batch Size | Max Length | Top-K | Algo | Epochs |
|---|---|---|---|---|---|---|
| NaiveGeneration | Qwen2.5 / GPT-4o-mini | – | $\infty$ | N/A | – | – |
| StandardRAG | Qwen2.5 / GPT-4o-mini | – | $\infty$ | 5 Chunks | – | – |
| GraphRAG | GPT-4o-mini | – | $\infty$ | 60 | – | – |
| LightRAG | GPT-4o-mini | – | $\infty$ | 60 | – | – |
| PathRAG | GPT-4o-mini | – | $\infty$ | 60 | – | – |
| HippoRAG2 | GPT-4o-mini | – | $\infty$ | 60 | – | – |
| HyperGraphRAG | GPT-4o-mini | – | $\infty$ | 60 | – | – |
| SFT | Qwen2.5 (7B) | 16 | 4096 | N/A | LoRA | 3 |
| R1 | Qwen2.5 (7B) | 128 | 4096 | N/A | GRPO | 3 |
| Search-R1 | Qwen2.5 (7B) | 128 | 4096 | 5 Chunks / Turn | GRPO | 6 |
| Search-R1-PPO | Qwen2.5 (7B) | 128 | 4096 | 5 Chunks / Turn | PPO | 10 |
| R1-Searcher | Qwen2.5 (7B) | 128 | 4096 | 5 Chunks / Turn | GRPO | 3 |
| Graph-R1 | Qwen2.5 (7B) | 128 | 4096 | 5 Chunks / Turn | GRPO | 3 |

Table 12: Hyperparameter settings in Graph-R1 setting.

