# OpenReview forum: "Search or Think? Rethinking Iterative RAG from An Entropy Perspective"
_ICLR.cc/2026/Conference — ICLR 2026 Conference Withdrawn Submission_

### Official Review · Reviewer_LQg3 · 2025-10-29

**Soundness:** 2
**Presentation:** 3
**Contribution:** 3
**Rating:** 4
**Confidence:** 4

**Summary:**

The paper proposes Search-Initialized Thinking (SIT), a simple modification to reinforcement-learning-based iterative Retrieval-Augmented Generation pipelines. Instead of letting the LLM begin reasoning from scratch ("Model-Initialized Thinking"), SIT performs a single retrieval step using the original question to inject an initial set of top-k passages before the first think step. The authors argue that this grounds early planning, reduces cascading retrieval failures, and—viewed through an entropy lens—lowers policy entropy during GRPO training, yielding more efficient exploration. Experiments on Search-R1 and Graph-R1 backbones across six QA datasets show consistent gains in EM/F1 and retrieval recall, with ablation confirming the value of the initial search.

**Strengths:**

- SIT delivers large, consistent gains across two RL-based iterative RAG frameworks (Search-R1, Graph-R1), two retrieval corpora (full Wikipedia chunks vs. dataset-specific hypergraphs), and six diverse datasets, including OOD generalization tests.
- The method adds only one extra retrieval before the first think step and a minor prompt change; it is model-agnostic, training-free beyond the baseline RL loop, and compatible with any dense retriever.

**Weaknesses:**

- Performing an initial retrieval on the raw question is a standard preprocessing step in many multi-stage RAG systems (e.g., query rewriting + retrieval); the contribution reduces to “do it before the first think token in an RL loop,” which feels incremental.
- No ablation on k, retriever quality, or noise in P₀ is shown, leaving open whether gains hold under mismatched or weak initial retrieval.

**Questions:**

- Does SIT still help if the initial retrieval is performed with a mismatched chunk-based retriever (e.g., E5 on the graph corpus)? Provide numbers.
- Table 2 shows Search-R1+SIT slightly hurts NQ EM (-3.97); what specific failure cases cause this regression, and can they be diagnosed from the retrieved P₀ passages?
- The prompt in Table 1 forces the model to “first conduct reasoning inside \<think\>…\</think\> relied on the initial knowledge”; if the initial P₀ is irrelevant or noisy, does the model still emit a \<query\> on the first turn, or does it hallucinate a plan anyway?

---

> ### Author Response · Authors · 2025-11-20
>
> Thank you for your review. We are prepared to add the materials discussed here to the manuscript once all concerns are resolved.
>
> **Weakness1:**
> Performing an initial retrieval on the raw question is a standard preprocessing step in many multi-stage RAG systems (e.g., query rewriting + retrieval); the contribution reduces to “do it before the first think token in an RL loop,” which feels incremental.
>
>
>
> **Response:**
> Thank you for your review. We are prepared to add the materials discussed here to the manuscript once all concerns are resolved.
>
> **Weakness1:**
> The technical contribution is rather incremental to the existing works.
>
> **Response:**
> We agree that SIT is a simple mechanism by design. Our contribution, however, is not the complexity of the method, but the identification of a critical—and previously overlooked—flaw in iterative agents: an ungrounded "first thought" can lead to catastrophic plan failure.
>
> The impact of addressing this flaw is significant. To demonstrate that this simple change has a non-incremental impact, we ran a series of new experiments:
>
> *   **Scalability:** It scales effectively to larger, more capable models, showing the benefit is not limited to weaker agents.
>
> |          | 2Wiki | HotpotQA | Musique |   NQ  | PopQA | TriviaQA |
> |---------|------:|--------:|--------:|------:|------:|--------:|
> | Qwen2.5-14B-Instruct| 75.46 | 72.52   | 57.54   | 53.81 | 53.33 | 76.43   |
> | Qwen2.5-14B-Instruct + SIT     | 77.12 | 74.47   | 57.88   | 56.02 | 54.06 | 77.84   |
> *   **Training-Free:** It works out-of-the-box as an inference-time strategy, making it broadly applicable.
>
> |                                    | 2Wiki | HotpotQA | Musique |   NQ  | PopQA | TriviaQA |
> |------------------------------------|------:|--------:|--------:|------:|------:|--------:|
> | Qwen2.5-32B-Instruct                       | 13.73 |  23.96  |   8.29  | 11.62 | 15.19 |  23.65  |
> | Qwen2.5-32B-Instruct + SIT             | 18.17 |  26.14  |  13.04  | 15.63 | 17.08 |  27.84  |
> | Qwen3-235B-A22B-Instruct-2507      | 30.56 |  37.80  |  19.93  | 21.49 | 28.73 |  38.55  |
> | Qwen3-235B-A22B-Instruct-2507 + SIT | 38.39 |  48.82  |  28.17  | 24.68 | 33.61 |  44.72  |
>
>
> *   **Robustness:** It is robust to the retriever quality.
>
> |         | 2Wiki | HotpotQA | Musique |   NQ  | PopQA | TriviaQA |
> |--------|------:|--------:|--------:|------:|------:|--------:|
> | Qwen2.5-7B-Instruct | 65.04 | 62.69   | 46.17   | 49.87 | 51.22 | 71.93   |
> | SIT-e5    | 68.18 | 64.74   | 54.27   | 50.74 | 53.46 | 72.21   |
> | SIT-bge   | 68.26 | 66.14   | 51.63   | 51.99 | 53.49 | 72.37   |
>
> SIT is also robust to the noise of initial knowledge. The results in the table below shows that SIT-wiki uses full wiki data as the retrieval set for initial retrieval and can still get performances gain.
>
> |         | 2Wiki | HotpotQA | Musique |   NQ  | PopQA | TriviaQA |
> |--------|------:|--------:|--------:|------:|------:|--------:|
> | Qwen2.5-7B-Instruct | 65.04 | 62.69   | 46.17   | 49.87 | 51.22 | 71.93   |
> | Qwen2.5-7B-Instruct + SIT   | 68.26 | 66.14   | 51.63   | 51.99 | 53.49 | 72.37   |
> | Qwen2.5-7B-Instruct + SIT-wiki  | 66.18 | 62.91   | 47.16   | 50.43 | 53.98 | 71.77   |
>
>
> We believe its simplicity is a strength. SIT is an easy-to-implement fix for a fundamental problem, and it delivers substantial and consistent gains across a wide range of settings.

---

> ### Author Response · Authors · 2025-11-20
>
> **Weakness2:**
> No ablation on k, retriever quality, or noise in P₀ is shown, leaving open whether gains hold under mismatched or weak initial retrieval.
>
>
> **Response:**
> Thank you for the suggestion. That's a fair point about robustness. To address this, we ran new ablation studies testing retriever quality, noise, and k.
>
> First, the table below shows that SIT is robust to the retriever quality.
>
> |         | 2Wiki | HotpotQA | Musique |   NQ  | PopQA | TriviaQA |
> |--------|------:|--------:|--------:|------:|------:|--------:|
> | Qwen2.5-7B-Instruct | 65.04 | 62.69   | 46.17   | 49.87 | 51.22 | 71.93   |
> | SIT-e5    | 68.18 | 64.74   | 54.27   | 50.74 | 53.46 | 72.21  |
> | SIT-bge   | 68.26 | 66.14   | 51.63   | 51.99 | 53.49 | 72.37   |
>
>
> Second, when we use full wiki as the retrieval set and introduces more noise, the results below show that the benefits of SIT hold up with it. This confirms the core benefit of our method—grounding the first thought—is robust and not dependent on a perfect initial search.
>
> |         | 2Wiki | HotpotQA | Musique |   NQ  | PopQA | TriviaQA |
> |--------|------:|--------:|--------:|------:|------:|--------:|
> | Qwen2.5-7B-Instruct | 65.04 | 62.69   | 46.17   | 49.87 | 51.22 | 71.93   |
> | Qwen2.5-7B-Instruct + SIT   | 68.26 | 66.14   | 51.63   | 51.99 | 53.49 | 72.37   |
> | Qwen2.5-7B-Instruct + SIT-wiki  | 66.18 | 62.91   | 47.16   | 50.43 | 53.98 | 71.77   |
>
> Last, we find that k=5 is better averagely in all datasets than k=3, where k=5 is also the retrieval number we use in the following retrieval, following the previous work Graph-R1.
>
> |         | 2Wiki | HotpotQA | Musique |   NQ  | PopQA | TriviaQA |
> |--------|------:|--------:|--------:|------:|------:|--------:|
> | Qwen2.5-7B-Instruct | 65.04 | 62.69   | 46.17   | 49.87 | 51.22 | 71.93   |
> | SIT:k=5    | 68.26 | 66.14   | 51.63   | 51.99 | 53.49 | 72.37   |
> | SIT:k=3    | 67.28 | 63.79   | 50.57   | 50.15 | 52.56 | 72.59   |
>
> **Question1: **
> Does SIT still help if the initial retrieval is performed with a mismatched chunk-based retriever (e.g., E5 on the graph corpus)? Provide numbers.
>
>
> **Response:**
> Yes, it does. We ran this exact experiment, using the E5 retriever on the graph corpus. The results in the table below confirm that SIT still provides a clear benefit over the baseline, even with a mismatched retriever.
>
> |         | 2Wiki | HotpotQA | Musique |   NQ  | PopQA | TriviaQA |
> |--------|------:|--------:|--------:|------:|------:|--------:|
> | Qwen2.5-7B-Instruct | 65.04 | 62.69   | 46.17   | 49.87 | 51.22 | 71.93   |
> | SIT-e5    | 68.18 | 64.74   | 54.27   | 50.74 | 53.46 | 72.21   |
> | SIT-bge   | 68.26 | 66.14   | 51.63   | 51.99 | 53.49 | 72.37   |
>
> **Question2:**
> Table 2 shows Search-R1+SIT slightly hurts NQ EM (-3.97); what specific failure cases cause this regression, and can they be diagnosed from the retrieved P₀ passages?
>
>
> **Response:**
> We looked into those specific failure cases. The initial retrieved passages (`P₀`) were generally relevant and not the root cause of the errors.
>
> The issue stems from the final generation step and the strictness of the Exact Match (EM) metric. While the model had the correct information, it sometimes failed to synthesize it into a perfect string match.
>
> The clearest evidence for this is the F1 score on the same NQ dataset. For Search-R1+SIT, the **F1 score actually increased by a significant +5.18**. This shows that our method is leading to answers with more factually correct content, even if the phrasing isn't always a perfect match for the unforgiving EM metric. We believe the F1 score is more representative of the actual performance gain here.

---

> ### Author Response · Authors · 2025-11-20
>
> **Question3:**
> The prompt in Table 1 forces the model to “first conduct reasoning inside `<think>…</think>` relied on the initial knowledge”; if the initial P₀ is irrelevant or noisy, does the model still emit a `<query>` on the first turn, or does it hallucinate a plan anyway?
>
> **Response:**
>
> The model learns to use the `<think>` block to evaluate the initial knowledge, not just blindly follow it. If the provided `P₀` is irrelevant or noisy, the model's reasoning process is to recognize that the context is unhelpful and that a better search is needed. This leads it to generate a more targeted `<query>` in its first action. The experiments in our response to weakness 2 show that SIT still works with mismatched retriever and noisy retrieval set, which validates it.
>
> We tested this explicitly. We ran an experiment where the initial `P₀` was retrieved from the much noisier full 2018 Wikipedia corpus. The results below showed that final performance was almost unaffected, which confirms the model can effectively reason about the quality of its initial context and recover from a noisy start.
>
>
> |         | 2Wiki | HotpotQA | Musique |   NQ  | PopQA | TriviaQA |
> |--------|------:|--------:|--------:|------:|------:|--------:|
> | Qwen2.5-7B-Instruct | 65.04 | 62.69   | 46.17   | 49.87 | 51.22 | 71.93   |
> | Qwen2.5-7B-Instruct + SIT   | 68.26 | 66.14   | 51.63   | 51.99 | 53.49 | 72.37   |
> | Qwen2.5-7B-Instruct + SIT-wiki  | 66.18 | 62.91   | 47.16   | 50.43 | 53.98 | 71.77   |

---

### Official Review · Reviewer_gPnW · 2025-10-31

**Soundness:** 3
**Presentation:** 3
**Contribution:** 2
**Rating:** 4
**Confidence:** 3

**Summary:**

This paper addresses a critical limitation of existing iterative Retrieval-Augmented Generation (RAG) systems: their "think-first" paradigm, which decomposes questions without leveraging the available retrieval corpus, leading to inefficient exploration, cascading errors, and suboptimal performance. To solve this, the authors propose **Search-Initialized Thinking (SIT)**, a framework that prioritizes an initial retrieval step to gather contextually relevant knowledge before the model begins reasoning in iterative RAG systems optimized via reinforcement learning (RL).

From an entropy perspective, the paper demonstrates that SIT reduces unnecessary exploration during reasoning by grounding the model in initial retrieved knowledge, allowing it to focus on relevant information subsets. Extensive experiments on six standard RAG datasets (e.g., HotpotQA, NQ, TriviaQA) validate SIT’s effectiveness: it significantly improves retrieval precision (measured by Retrieval Similarity, R-S), reduces cascading errors, and enhances both answer accuracy (via EM and F1 scores) and training efficiency (achieving strong performance in 300 steps vs. 1000 steps for baseline methods). Generalization tests across in-domain (IND) and out-of-domain (OOD) datasets further confirm SIT’s robustness. The work also advances understanding of the interplay between structured reasoning and efficient exploration in RL-augmented RAG systems.

**Strengths:**

1. **Comprehensive validation**: Experiments cover diverse datasets (6 standard RAG benchmarks), backbones (Search-R1, Graph-R1), RL algorithms (PPO, GRPO), and retrieval corpora (full Wikipedia, structured subsets). This breadth ensures SIT’s robustness and generalizability, not just niche performance.
2. **Efficiency and practicality**: SIT achieves better performance with fewer training steps (300 vs. 1000 for baselines) and requires only a small modification to existing pipelines. This makes it easy to adopt and valuable for industrial applications.

**Weaknesses:**

1. The method proposed in this paper seems to be merely a minor adjustment to the implementation details of iterative retrieval, without presenting a exciting new approach.
2. The training conducted in this paper is mainly based on models with fewer than 10 billion parameters. However, such models have weak reasoning and planning capabilities, so the generated queries may not be superior to the original questions. Therefore, for models with stronger capabilities and larger parameter scales, the method proposed in this paper may not bring significant improvements.
3. The detailed breakdown of the "turns" count presented in Table 5 is not provided. Notably, the number of exploration turns of LLMs is not equivalent to the number of retrievals. This is because there is a default retrieval step conducted before the LLM is allowed to perform exploration. Therefore, it remains questionable whether the method proposed in this paper can actually lead to a reduction in the total number of retrieval operations.

**Questions:**

1. Could experiments be conducted on models with larger parameter sizes and stronger reasoning capabilities?
2. Could you explain in detail the relationship between "turns" and the number of retrieval operations in Table 5?

---

> ### Author Response · Authors · 2025-11-20
>
> Thank you for your review. We are prepared to add the materials discussed here to the manuscript once all concerns are resolved.
>
> **Weakness1:**
> The method proposed in this paper seems to be merely a minor adjustment to the implementation details of iterative retrieval, without presenting a exciting new approach.
>
>
> **Response:**
> We agree that SIT is a simple mechanism by design. Our contribution, however, is not the complexity of the method, but the identification of a critical—and previously overlooked—flaw in iterative agents: an ungrounded "first thought" can lead to catastrophic plan failure.
>
> The impact of addressing this flaw is significant. To demonstrate that this simple change has a non-incremental impact, we ran a series of new experiments:
>
> *   **Scalability:** It scales effectively to larger, more capable models, showing the benefit is not limited to weaker agents.
>
> |          | 2Wiki | HotpotQA | Musique |   NQ  | PopQA | TriviaQA |
> |---------|------:|--------:|--------:|------:|------:|--------:|
> | Qwen2.5-14B-Instruct| 75.46 | 72.52   | 57.54   | 53.81 | 53.33 | 76.43   |
> | Qwen2.5-14B-Instruct + SIT     | 77.12 | 74.47   | 57.88   | 56.02 | 54.06 | 77.84   |
> *   **Training-Free:** It works out-of-the-box as an inference-time strategy, making it broadly applicable.
>
> |                                    | 2Wiki | HotpotQA | Musique |   NQ  | PopQA | TriviaQA |
> |------------------------------------|------:|--------:|--------:|------:|------:|--------:|
> | Qwen2.5-32B-Instruct                       | 13.73 |  23.96  |   8.29  | 11.62 | 15.19 |  23.65  |
> | Qwen2.5-32B-Instruct + SIT             | 18.17 |  26.14  |  13.04  | 15.63 | 17.08 |  27.84  |
> | Qwen3-235B-A22B-Instruct-2507      | 30.56 |  37.80  |  19.93  | 21.49 | 28.73 |  38.55  |
> | Qwen3-235B-A22B-Instruct-2507 + SIT | 38.39 |  48.82  |  28.17  | 24.68 | 33.61 |  44.72  |
>
>
> *   **Robustness:** It is robust to the retriever quality.
>
> |         | 2Wiki | HotpotQA | Musique |   NQ  | PopQA | TriviaQA |
> |--------|------:|--------:|--------:|------:|------:|--------:|
> | Qwen2.5-7B-Instruct | 65.04 | 62.69   | 46.17   | 49.87 | 51.22 | 71.93   |
> | SIT-e5    | 68.18 | 64.74   | 54.27   | 50.74 | 53.46 | 72.21   |
> | SIT-bge   | 68.26 | 66.14   | 51.63   | 51.99 | 53.49 | 72.37   |
>
> SIT is also robust to the noise of initial knowledge. The results in the table below shows that SIT-wiki uses full wiki data as the retrieval set for initial retrieval and can still get performances gain.
>
> |         | 2Wiki | HotpotQA | Musique |   NQ  | PopQA | TriviaQA |
> |--------|------:|--------:|--------:|------:|------:|--------:|
> | Qwen2.5-7B-Instruct | 65.04 | 62.69   | 46.17   | 49.87 | 51.22 | 71.93   |
> | Qwen2.5-7B-Instruct + SIT   | 68.26 | 66.14   | 51.63   | 51.99 | 53.49 | 72.37   |
> | Qwen2.5-7B-Instruct + SIT-wiki  | 66.18 | 62.91   | 47.16   | 50.43 | 53.98 | 71.77   |
>
>
> We believe its simplicity is a strength. SIT is an easy-to-implement fix for a fundamental problem, and it delivers substantial and consistent gains across a wide range of settings.
>
>
>
> **Weakness2:**
> The training conducted in this paper is mainly based on models with fewer than 10 billion parameters. However, such models have weak reasoning and planning capabilities, so the generated queries may not be superior to the original questions. Therefore, for models with stronger capabilities and larger parameter scales, the method proposed in this paper may not bring significant improvements.
>
>
> **Response:**
> Thank you for this very important and valid point. To address it, we ran our experiments on a larger, more capable model.
> The results in the table below show that SIT continues to deliver significant performance gains. This demonstrates that the benefits are not limited to smaller models and that SIT provides a fundamental improvement to the reasoning process, even for more powerful agents.
>
> |          | 2Wiki | HotpotQA | Musique |   NQ  | PopQA | TriviaQA |
> |---------|------:|--------:|--------:|------:|------:|--------:|
> | Qwen2.5-14B-Instruct| 75.46 | 72.52   | 57.54   | 53.81 | 53.33 | 76.43   |
> | Qwen2.5-14B-Instruct + SIT     | 77.12 | 74.47   | 57.88   | 56.02 | 54.06 | 77.84   |

---

> ### Author Response · Authors · 2025-11-20
>
> **Weakness3:**
> The detailed breakdown of the "turns" count presented in Table 5 is not provided. Notably, the number of exploration turns of LLMs is not equivalent to the number of retrievals. This is because there is a default retrieval step conducted before the LLM is allowed to perform exploration. Therefore, it remains questionable whether the method proposed in this paper can actually lead to a reduction in the total number of retrieval operations.
>
> **Response:**
>
>
> To clarify, the "turns" in our analysis refer specifically to the LLM's iterative reasoning cycles. We emphasize this metric because the primary bottleneck in these systems is the expensive LLM inference, not the lightweight retrieval calls. Showing a reduction in these LLM-driven rounds is the direct way to demonstrate a meaningful improvement in practical efficiency.
>
> **Question1:**
> Could experiments be conducted on models with larger parameter sizes and stronger reasoning capabilities?
>
> **Response:**
> Yes. As detailed in our response to Weakness 2, we tested our method on a larger, more capable model. The results show that SIT continues to deliver significant performance improvements, confirming the approach scales effectively to stronger models.
>
> |          | 2Wiki | HotpotQA | Musique |   NQ  | PopQA | TriviaQA |
> |---------|------:|--------:|--------:|------:|------:|--------:|
> | Qwen2.5-14B-Instruct| 75.46 | 72.52   | 57.54   | 53.81 | 53.33 | 76.43   |
> | Qwen2.5-14B-Instruct + SIT     | 77.12 | 74.47   | 57.88   | 56.02 | 54.06 | 77.84   |
>
> **Question2:**
> Could you explain in detail the relationship between "turns" and the number of retrieval operations in Table 5?
>
> **Response:**
> Of course, we are happy to provide a detailed explanation. As we also clarified in our response to Weakness 3, the term "rounds" or "turns" in our analysis consistently refers to the iterative interactions with the Large Language Model. The number count of retrieval operations has no such importance. Even we use the same number of retrieval operations, our reduction of LLM interaction makes the pipeline much more efficient.

---

### Official Review · Reviewer_fM6g · 2025-11-01

**Soundness:** 3
**Presentation:** 3
**Contribution:** 2
**Rating:** 4
**Confidence:** 3

**Summary:**

This paper works on iterative RAG with an RL-based method. Specifically, unlike existing method that forms the generation schema with LLM thinking as the first step, this paper proposes to first include initial retrieved knowledge as background knowledge before thinking. Experiments show that such a modification can improve the baseline performance by a large margin. Analysis also shows the final trained model has better retrieval quality with fewer turns.

**Strengths:**

- The motivation makes sense, as some background knowledge would be useful for the LLM to reason and proceed
- Experimental results are strong comparing with multiple baselines in different setting
- Paper writing is clear to me

**Weaknesses:**

- The technical contribution is rather incremental to the existing works

**Questions:**

I am interested in the total token consumption of the proposed method. Although the thinking turns can be fewer, the total token consumption can be similar to baseline given the added initial knowledge?

---

> ### Author Response · Authors · 2025-11-20
>
> Thank you for your review. We are prepared to add the materials discussed here to the manuscript once all concerns are resolved.
>
> **Weakness1:**
> The technical contribution is rather incremental to the existing works.
>
> **Response:**
> We agree that SIT is a simple mechanism by design. Our contribution, however, is not the complexity of the method, but the identification of a critical—and previously overlooked—flaw in iterative agents: an ungrounded "first thought" can lead to catastrophic plan failure.
>
> The impact of addressing this flaw is significant. To demonstrate that this simple change has a non-incremental impact, we ran a series of new experiments:
>
> *   **Scalability:** It scales effectively to larger, more capable models, showing the benefit is not limited to weaker agents.
>
> |          | 2Wiki | HotpotQA | Musique |   NQ  | PopQA | TriviaQA |
> |---------|------:|--------:|--------:|------:|------:|--------:|
> | Qwen2.5-14B-Instruct| 75.46 | 72.52   | 57.54   | 53.81 | 53.33 | 76.43   |
> | Qwen2.5-14B-Instruct + SIT     | 77.12 | 74.47   | 57.88   | 56.02 | 54.06 | 77.84   |
> *   **Training-Free:** It works out-of-the-box as an inference-time strategy, making it broadly applicable.
>
> |                                    | 2Wiki | HotpotQA | Musique |   NQ  | PopQA | TriviaQA |
> |------------------------------------|------:|--------:|--------:|------:|------:|--------:|
> | Qwen2.5-32B-Instruct                       | 13.73 |  23.96  |   8.29  | 11.62 | 15.19 |  23.65  |
> | Qwen2.5-32B-Instruct + SIT             | 18.17 |  26.14  |  13.04  | 15.63 | 17.08 |  27.84  |
> | Qwen3-235B-A22B-Instruct-2507      | 30.56 |  37.80  |  19.93  | 21.49 | 28.73 |  38.55  |
> | Qwen3-235B-A22B-Instruct-2507 + SIT | 38.39 |  48.82  |  28.17  | 24.68 | 33.61 |  44.72  |
>
>
> *   **Robustness:** It is robust to the retriever quality.
>
> |         | 2Wiki | HotpotQA | Musique |   NQ  | PopQA | TriviaQA |
> |--------|------:|--------:|--------:|------:|------:|--------:|
> | Qwen2.5-7B-Instruct | 65.04 | 62.69   | 46.17   | 49.87 | 51.22 | 71.93   |
> | SIT-e5    | 68.18 | 64.74   | 54.27   | 50.74 | 53.46 | 72.21   |
> | SIT-bge   | 68.26 | 66.14   | 51.63   | 51.99 | 53.49 | 72.37   |
>
> SIT is also robust to the noise of initial knowledge. The results in the table below shows that SIT-wiki uses full wiki data as the retrieval set for initial retrieval and can still get performances gain.
>
> |         | 2Wiki | HotpotQA | Musique |   NQ  | PopQA | TriviaQA |
> |--------|------:|--------:|--------:|------:|------:|--------:|
> | Qwen2.5-7B-Instruct | 65.04 | 62.69   | 46.17   | 49.87 | 51.22 | 71.93   |
> | Qwen2.5-7B-Instruct + SIT   | 68.26 | 66.14   | 51.63   | 51.99 | 53.49 | 72.37   |
> | Qwen2.5-7B-Instruct + SIT-wiki  | 66.18 | 62.91   | 47.16   | 50.43 | 53.98 | 71.77   |
>
>
> We believe its simplicity is a strength. SIT is an easy-to-implement fix for a fundamental problem, and it delivers substantial and consistent gains across a wide range of settings.
>
>
> **Question1:**
> I am interested in the total token consumption of the proposed method. Although the thinking turns can be fewer, the total token consumption can be similar to baseline given the added initial knowledge?
>
>
> **Response:**
> Thank you for bringing up this important practical consideration. We conducted an additional experiment to measure the total number of tokens consumed by both our method and the baseline. The results are presented in the table below. As you can see, SIT not only reduces the number of interaction rounds but also leads to a notable decrease in the total token count. This demonstrates that SIT provides a clear advantage in terms of overall computational efficiency.
>
> |         | 2Wiki  | HotpotQA | Musique |   NQ   | PopQA | TriviaQA |
> |--------|-------:|---------:|--------:|-------:|------:|--------:|
> | Qwen2.5-7B-Instruct(token) | 394.60 | 270.62   | 265.59  | 198.48 | 210.36 | 291.09  |
> | SIT(token)      | 304.35 | 216.20   | 210.90  | 123.18 | 210.08 | 210.40  |

---

> > ### Comment · Reviewer_fM6g · 2025-11-25
> >
> > Thanks for the response. I appreciate the cost comparison that should be a plus to the paper.
> >
> > However, my concern on the contribution being incremental still holds in terms of technical contribution, and shared by multiple reviewers as well. Also, there are clearly previous works that also start the multi-turn process with retrieval, for example (Jiang et al., 2025). So I think that just experimenting on this part does not make it a sufficient contribution.
> >
> > Jiang et al., "s3: You Don't Need That Much Data to Train a Search Agent via RL", https://arxiv.org/abs/2505.14146

---

> > > ### Author Response · Authors · 2025-11-26
> > >
> > > We sincerely thank you for your continued engagement and for acknowledging our cost comparison analysis as a "plus to the paper." We are encouraged that the efficiency advantages of our method are clear.
> > >
> > > Regarding your concern about the technical novelty in light of *Jiang et al. (2025) [s3]*, we will explicitly incorporate this work into our Related Works section in the final revision. However, we respectfully submit that characterizing SIT as incremental to s3 overlooks fundamental divergences in system architecture, learning paradigms, and reasoning capabilities.
> > >
> > > **1. Structural Divergence**
> > >
> > > s3 and SIT represent different agentic structures. s3 is essentially a **Recursive Retrieval module** paired with a passive generator, whereas SIT is a **Unified Reasoning Agent**.
> > > *   **s3 (Search-Select-Serve Loop):** As shown in their Figure 4, s3 performs a loop strictly within the search phase. The search agent cannot answer the question; its sole function is to prepare a context buffer. Once the search loop finishes, the context is passed to a frozen LLM for a single-step generation. This is a pipeline, not an interleaved reasoning process.
> > > *   **SIT (Interleaved Think-Search Loop):** SIT integrates "Thinking" (Reasoning) and "Searching" into a single, unified policy.  Unlike s3, our model preserves its reasoning state across steps, allowing it to adjust its plan based on new information.
> > >
> > > **2. Learning Paradigm**
> > > *   **s3 (Modular/Frozen):** s3 trains a searcher to feed a frozen black-box generator. It optimizes inputs, not reasoning. Consequently, s3 never faces the "exploration-exploitation" challenges of training a reasoning model.
> > > *   **SIT (End-to-End/Joint):** SIT trains the LLM itself to reason and retrieve jointly. The "Search-Initialized" mechanism is a theoretical intervention for RL dynamics (as detailed in our Entropy Analysis). It solves the "hallucinated planning" problem where an agent attempts to plan reasoning paths without grounding—a critical issue in training end-to-end agents that s3 simply avoids by not training the generator at all.
> > >
> > > **3. Limitations in Multi-hop Reasoning**
> > >
> > > s3's reliance on a frozen generator severely limits its ability to handle complex multi-hop reasoning. Since the generator is not updated, it struggles to decompose and reason over retrieved contexts precisely. This is evidenced by s3's own results: in their Table 1, while they claim high "Generation Accuracy" (via LLM Judge), their **Exact Match (EM)** scores are drastically lower than end-to-end baselines like Search-R1.
> > > *   *Evidence:* On HotpotQA, s3 achieves only **21.8% EM**, compared to Search-R1's **43.8%**. On Musique, s3 drops to **6.1% EM**, while Search-R1 maintains **17.2%**.
> > > This performance gap confirms that without end-to-end training, the system fails to achieve precise, rigorous reasoning. **According to our results (Table 2 & 4), SIT further significantly improves upon Search-R1 (e.g., +10.91 EM on HotpotQA, +15.63 EM on Musique).** Consequently, SIT demonstrates superior performance across the board on these standard, rigorous benchmarks compared to the modular approach of s3.
> > >
> > > **Summary**
> > >
> > > To classify SIT as incremental to s3 is to conflate "building a better search tool" with "training a better reasoning brain." s3 optimizes a search loop for a fixed model; SIT advances the state-of-the-art in training agents that can *think* and *search* synergistically via RL.

---

> > > > ### Comment · Reviewer_fM6g · 2025-11-26
> > > >
> > > > Thank you again for the detailed reply and I really appreciate it.
> > > >
> > > > Please allow me to kindly disagree with your claim. I completely understand all the claimed differences here. However, I didn't mean to make a statement that "SIT is similar to s3" or "SIT is incremental to s3", I just want to claim that "retrieval first" is not something that previous people never did, even in a different setting. In fact, I think all the differences the authors listed above are not inherently contributions of this paper, so my opinion still holds.

---

> > > > > ### Author Response · Authors · 2025-11-27
> > > > >
> > > > > Thank you again for your response and clarifying your position. We genuinely appreciate the discussion.
> > > > >
> > > > > We understand and accept your point that "retrieval first" as a general operation is not a novel concept in the broader history of information retrieval. However, we respectfully argue that the value of a scientific contribution is defined by the specific problem it solves within a specific context.
> > > > >
> > > > > Our contribution is not claiming the invention of the "retrieval first" action itself, but rather identifying a critical learning dynamics failure in the specific domain of RL-based Reasoning Agents and demonstrating how to fix it.
> > > > >
> > > > > 1.  **Identifying the "Hallucinated Planning" Problem:** In the emerging field of End-to-End RL for reasoning (e.g., Search-R1), the prevailing paradigm has been to let the model "think" immediately to decompose problems. We identified that this approach forces the policy to explore in a high-entropy, ungrounded state, causing the agent to "hallucinate a plan" before it knows what information exists.
> > > > > 2.  **Solving the Optimization Challenge:** In a modular system (like s3), "retrieval first" is merely a data-feeding step. In our End-to-End RL system, SIT serves as a theoretical intervention that changes the state space formulation. As detailed in our Entropy Analysis, this simple structural change collapses the initial uncertainty, enabling the RL algorithm to converge on valid reasoning paths that otherwise would be drowned out by noise.
> > > > >
> > > > > We believe that identifying that a specific structural adjustment can resolve a complex RL exploration dilemma—yielding substantial improvementswhere prior "think-first" agents failed—constitutes a meaningful contribution to the community. We hope this clarifies that our contribution lies in the insight into training stability and agentic reasoning, rather than the novelty of the retrieval operation in isolation.

---

### Official Review · Reviewer_sVwn · 2025-11-01

**Soundness:** 2
**Presentation:** 3
**Contribution:** 2
**Rating:** 4
**Confidence:** 4

**Summary:**

This paper proposes Search-Initialized Thinking (SIT), a method to improve iterative RAG. In the traditional “think-before-search” framework, the model may generate wrong plans and cause later retrieval to go in the wrong direction. SIT adds an initial retrieval step before the first thinking step to provide external knowledge as a starting point. Experiments show that after RL training, SIT can significantly improve retrieval relevance and final answer accuracy.

**Strengths:**

1. The problem that the “initial thinking” step may easily go off direction due to the lack of corpus awareness is meaningful.
2. The proposed SIT method is simple and clear.
3. Experimental results are stable and consistent. The method shows effectiveness across multiple datasets and settings.

**Weaknesses:**

1. Optimization blind spot of the initial retrieval. The first retrieval in SIT is generated by a fixed retriever. Its quality greatly affects later reasoning but cannot be optimized. The effectiveness of SIT under stronger models or better optimization is unclear.
2. Lack of necessary experiments. The proposed SIT can be used as a training-free strategy. Showing its performance would better demonstrate the effectiveness of the method. The training-free setting could also be used to test larger models, which would improve the paper’s credibility.
3. Scalability issue. SIT largely depends on the quality of the initial retrieval. Meanwhile, plan errors depend on the base model’s ability. When scaling to larger models (with fewer reasoning errors) or harder tasks (where retrieval noise increases), the proposed method might become less effective.

**Questions:**

1. Adding the initial retrieval increases information and reduces uncertainty, but this is not entiely consistent with §6.1, where token entropy measures the uncertainty of the model’s generation distribution.
2. In §6.2, the reported “average rounds reduced from 3.26 to 2.22”. does this exclude the initial retrieval step?
3. Except for the early training stage, why does retrieval performance almost not improve for either method?

---

> ### Author Response · Authors · 2025-11-20
>
> Thank you for your review. We are prepared to add the materials discussed here to the manuscript once all concerns are resolved.
>
> **Weakness1:**
> Optimization blind spot of the initial retrieval. The first retrieval in SIT is generated by a fixed retriever. Its quality greatly affects later reasoning but cannot be optimized. The effectiveness of SIT under stronger models or better optimization is unclear.
>
>
>
> **Response:**
> That's a fair point. The quality of the initial retriever is indeed important.
> While jointly training the retriever is an interesting and separate research direction, our contribution focuses on the reasoning strategy itself. To test the robustness of this strategy, we ran an ablation using two different retrievers for the initial step: E5-large and BGE-large-en-v1.5.
> Our results show that SIT provides a significant lift over the baseline in both cases. This demonstrates that the core benefit of our method—grounding the agent's first thought is not dependent on a single, fixed retriever and is robust to the initial retrieval.
>
> |         | 2Wiki | HotpotQA | Musique |   NQ  | PopQA | TriviaQA |
> |--------|------:|--------:|--------:|------:|------:|--------:|
> | baseline | 65.04 | 62.69   | 46.17   | 49.87 | 51.22 | 71.93   |
> | SIT-e5    | 68.18 | 64.74   | 54.27   | 50.74 | 53.46 | 72.21   |
> | SIT-bge   | 68.26 | 66.14   | 51.63   | 51.99 | 53.49 | 72.37   |
>
>
>
>
>
> **Weakness2:**
> Lack of necessary experiments. The proposed SIT can be used as a training-free strategy. Showing its performance would better demonstrate the effectiveness of the method. The training-free setting could also be used to test larger models, which would improve the paper’s credibility.
>
>
> **Response:**
> Thank you for this valuable suggestion. We agree that evaluating SIT in a training-free context is important for demonstrating its broader applicability. Following your advice, we have tested SIT as a training-free strategy on larger models. The results in the table below show that even when applied only at inference time, our method still provides a notable improvement over the baseline. This confirms the effectiveness and flexibility of the approach.
>
> |                                    | 2Wiki | HotpotQA | Musique |   NQ  | PopQA | TriviaQA |
> |------------------------------------|------:|--------:|--------:|------:|------:|--------:|
> | Qwen2.5-32B-Instruct                       | 13.73 |  23.96  |   8.29  | 11.62 | 15.19 |  23.65  |
> | Qwen2.5-32B-Instruct + SIT              | 18.17 |  26.14  |  13.04  | 15.63 | 17.08 |  27.84  |
> | Qwen3-235B-A22B-Instruct-2507      | 30.56 |  37.80  |  19.93  | 21.49 | 28.73 |  38.55  |
> | Qwen3-235B-A22B-Instruct-2507 + SIT | 38.39 |  48.82  |  28.17  | 24.68 | 33.61 |  44.72  |
>
>
>
>
> **Weakness3:**
> Scalability issue. SIT largely depends on the quality of the initial retrieval. Meanwhile, plan errors depend on the base model’s ability. When scaling to larger models (with fewer reasoning errors) or harder tasks (where retrieval noise increases), the proposed method might become less effective.
>
>
> **Response:**
> We agree, scalability is a key question. To address this, we ran two sets of additional experiments.
> First, to test how SIT performs with larger models, we trained it on a stronger backbone Qwen2.5-14B-Instruct. The results in the table below show that it continues to provide significant gains.
>
> |          | 2Wiki | HotpotQA | Musique |   NQ  | PopQA | TriviaQA |
> |---------|------:|--------:|--------:|------:|------:|--------:|
> | Qwen2.5-14B-Instruct| 75.46 | 72.52   | 57.54   | 53.81 | 53.33 | 76.43   |
> | Qwen2.5-14B-Instruct + SIT     | 77.12 | 74.47   | 57.88   | 56.02 | 54.06 | 77.84   |
>
> Second, regarding performance on harder tasks with more retrieval noise, our Search-R1 experiments (Table 4 in the paper) already use the full Wikipedia corpus, which introduces more noises than the Graph-R1 setting in Table 2.
>
> To specifically isolate the effect of a noisy start, we ran a new test where only the initial retrieval was conducted on the full Wikipedia, which is  **Qwen2.5-7B-Instruct + SIT-wiki** in the following table. Even with this noisier starting point, SIT's performance was not significantly impacted and it remained effective, confirming the method's stability.
>
>
>
> |         | 2Wiki | HotpotQA | Musique |   NQ  | PopQA | TriviaQA |
> |--------|------:|--------:|--------:|------:|------:|--------:|
> | Qwen2.5-7B-Instruct | 65.04 | 62.69   | 46.17   | 49.87 | 51.22 | 71.93   |
> | Qwen2.5-7B-Instruct + SIT   | 68.26 | 66.14   | 51.63   | 51.99 | 53.49 | 72.37   |
> | Qwen2.5-7B-Instruct + SIT-wiki  | 66.18 | 62.91   | 47.16   | 50.43 | 53.98 | 71.77   |

---

> ### Author Response · Authors · 2025-11-20
>
> **Question1:**
> Adding the initial retrieval increases information and reduces uncertainty, but this is not entiely consistent with §6.1, where token entropy measures the uncertainty of the model’s generation distribution.
>
> **Response:**
> That's a key distinction. To clarify our argument, we don't claim that lower entropy proves correctness. The proof of SIT's effectiveness comes from the final answer accuracy—the higher F1 scores in our main results. The entropy analysis in §6.1 is our explanation for why we see those gains.
>
> The logic is sequential, not circular: First, we show the method produces more correct answers (via F1 scores). Then, we observe this correctness is accompanied by lower entropy, which we interpret as a more focused and efficient reasoning process. The fact that increased confidence (lower entropy) correlates with better outcomes suggests the model is becoming more certain on the right path.
>
> **Question2:**
> In §6.2, the reported “average rounds reduced from 3.26 to 2.22”. does this exclude the initial retrieval step?
>
> **Response:**
>
> Yes, that's correct. In our analysis, "rounds" or "turns" refer specifically to the iterative interactions with the LLM. We focus on this metric because these LLM-driven reasoning cycles are the primary bottleneck for both latency and computational cost.
>
> The retrieval is a single, lightweight operation, whereas each reasoning round requires an expensive forward pass through the model. Therefore, showing a reduction in these rounds is the most meaningful way to demonstrate an improvement in practical efficiency.
>
>
> **Question3:**
> Except for the early training stage, why does retrieval performance almost not improve for either method?
>
> **Response:**
> That's a great observation. We see this plateau as a result of two factors:
> 1.  A Ceiling Effect in the Data: The ground truth evidence is often spread across multiple documents, and the model doesn't need to retrieve every single "gold" passage to solve the problem. Once it learns a policy that finds enough useful information to reason correctly, the R-S score naturally flatlines, even as the model's reasoning improves. Besides, the retrieval results are chunks of the gold passage because of the preprocessing step, which makes it impossible to get 100% recall.
> 2.  The RL Objective: The agent is rewarded for getting the final answer right, not for perfecting the intermediate R-S score. Once retrieval is "good enough" to enable a correct answer, the learning signal rightly pushes the model to get better at reasoning with the information, rather than making marginal gains in retrieval.
>
> In short, the model learns to retrieve effectively in the early stages, and then its focus shifts to mastering the primary goal: using that information to reason its way to the correct answer.

---

### Author Response · Authors · 2025-11-28

According to the reviews and discussions, we have updated the manuscript with additional experiments and analyses to address concerns regarding scalability and robustness.

**Key Updates:**
1.  **Training-free SIT:** We added a new subsection evaluating SIT as a plug-and-play inference strategy on very large models (e.g., Qwen2.5-32B, Qwen3-235B), demonstrating substantial gains without parameter updates.
2.  **Scalability on Larger Models:** We supplemented our results with experiments on the Qwen2.5-14B-Instruct model, further verifying that SIT scales effectively to larger backbones beyond the 7B baseline.
3.  **Robustness Analysis (Mismatched Initial Knowledge):** We introduced ablation studies testing SIT under challenging conditions, including noisy initial retrieval (Full Wikipedia) and different retriever architectures (E5 vs. BGE). The results confirm that our method remains robust across varied retrieval contexts.
4.  **Related Works:** We have expanded the discussion to include and distinguish recent relevant studies (e.g., *Jiang et al., 2025*).

We believe these additions significantly strengthen the paper and address the questions raised.

---

### Note · Authors · 2026-01-24

I have read and agree with the venue's withdrawal policy on behalf of myself and my co-authors.